# Convolutional Neural Networks for Sea Surface Data Assimilation in Operational Ocean Models: Test Case in the Gulf of Mexico

Olmo Zavala-Romero[1,2], Alexandra Bozec[2], Eric P. Chassignet[2], and Jose R. Miranda[1,2]

[1]Department of Scientific Computing, Florida State University, Tallahassee, FL 32306 USA
[2]Center for Ocean-Atmospheric Prediction Studies, Florida State University, Tallahassee, FL 32306 USA

**Correspondence:** Olmo Zavala-Romero (osz09@fsu.edu)

**Abstract.** Deep learning models have demonstrated remarkable success in fields such as language processing and computer vision, routinely employed for tasks like language translation, image classification, and anomaly detection. Recent advancements in ocean sciences, particularly in data assimilation (DA), suggest that machine learning can emulate dynamical models, replace traditional DA steps to expedite processes, or serve as hybrid surrogate models to enhance forecasts. However, these studies often rely on ocean models of intermediate complexity, which involve significant simplifications that present challenges when transitioning to full-scale operational ocean models. This work explores the application of Convolutional Neural Networks (CNNs) in data assimilation within the context of the Hybrid Coordinate Ocean Model (HYCOM) in the Gulf of Mexico. The CNNs are trained to correct model errors from a two-year, high-resolution (1/25°) HYCOM dataset, assimilated using the Tendral Statistical Interpolation System (T-SIS). The CNNs are trained to replicate the increments generated by the TSIS data assimilation package, aiming to correct model forecasts of sea surface temperature (SST) and sea surface height (SSH). The inputs to the CNNs include real satellite observations of SST from the Group for High Resolution Sea Surface Temperature (GHRSST), along-track altimeter SSH observations (ADT), the model background state (previous forecast), and the innovations (differences between observations and background). We assess the performance of the CNNs across five controlled experiments, designed to provide insights into their application in environments governed by full primitive equations, real observations, and complex topographies. The experiments focus on evaluating: 1) the architecture and complexity of the CNNs, 2) the type and quantity of observations, 3) the type and number of assimilated fields, 4) the impact of training window size, and 5) the influence of coastal boundaries. Our findings reveal significant correlations between the chosen training window size—a factor not commonly examined—and the CNNs' ability to assimilate observations effectively. We also establish a clear link between the CNNs' architecture and complexity and their overall performance.

This research uses artificial intelligence to enhance ocean forecasting in the Gulf of Mexico. By using Convolutional Neural Networks, the study improves predictions of sea temperatures and heights by integrating real satellite data with existing models. Through five comprehensive experiments, the team found that the amount of training data and the design of the neural networks significantly affect accuracy. These insights pave the way for faster, more reliable ocean models, benefiting environmental monitoring and maritime operations.

# 1  Introduction

Assimilating diverse observations into operational ocean models presents significant challenges, primarily due to the computational demands and complexities associated with traditional methods like Four-Dimensional Variational Data Assimilation (4DVar) or variations of the ensemble kalman filter (EnKF). These methods, while robust, require substantial computational resources and time, 4DVar in the integration of the adjoint model and EnKF in the integration of the physcan model itself. The data assimilation process can be particularly time consuming when dealing with heterogeneous and high-volume datasets, which are becoming more common in oceanographic research. Machine learning methods, on the other hand, offer a promising alternative that could potentially accelerate this assimilation process.

Recent works in ocean sciences explore the feasibility and effectiveness of using techniques such as neural networks (NNs) to improve ocean models. For example, it has been explored how the entire variational data assimilation system could be substituted with machine learning-based approaches (Geer, 2021; Boukabara et al., 2019; Dong et al., 2022). While this approach is still maturing, there is considerable interest in using machine learning to enhance existing data assimilation systems.

Additionally, machine learning methods have been applied to specific components of data assimilation systems in ocean models. For instance, it has been discussed how neural networks can be used for fast emulation of forward models, which are crucial for direct assimilation of satellite measurements in ocean models Krasnopolsky (2013). Furthermore, ML observation operators have been developed to improve the assimilation of surface observations such as sea surface temperature and ocean surface elevation (Guinehut et al., 2004).

This work investigates the use of CNNs to assimilate sea surface height and sea surface temperature observations with the HYbrid Coordinate Ocean Model (HYCOM). The CNNs are trained to correct the model error from a 1/25 resolution two-year-long data assimilated HYCOM run with the Tendral Statistical Interpolation System (T-SIS) as the assimilation package. The performance of the CNNs is studied through five controlled experiments that provide intuition on how to apply them in settings with full primitive equations, real observations, and complex topographies.

The experiments evaluate the architecture and complexity of the CNN, the type and number of observations, the type and number of assimilated fields, the response to the training window size, and the effects of the coastline. Our results show strong correlations between the window size selected to train the CNN, which is not commonly evaluated, and the ability of the CNN to assimilate the observations. Similarly, we found a clear relationship between the complexity of the chosen CNN and its overall performance.

Sections 2 and 2.1 provide a small overview of the HYCOM model and the T-SIS assimilation system. Section 2.2 provides an introduction to CNNs and the U-net architecture. Section 3 describes the controlled experiments using CNNs, section 4 describes the results, the generalization tests performed, and the performance comparison with T-SIS. We end with conclusions and final remarks in Section 5.

## 2 The Hybrid Coordinate System Ocean Model (HYCOM)

The HYbrid Coordinate Ocean Model (HYCOM) is a state-of-the-art multi-layer ocean model (Bleck, 2002; Chassignet et al., 2003, 2007, 2009; Chin et al., 1999). A key feature of HYCOM is the use of a hybrid vertical coordinate. While the horizontal coordinates are typically Cartesian, the vertical coordinate need not be restricted to represent the vertical distance from a specified origin, the so-called "z-coordinate". In various parts of an ocean basin, the layer flow may be driven more strongly by different processes, which in turn gives preference to the use of a more suitable vertical coordinate. In the open stratified ocean, for example, the ocean flow typically follows along layers of constant potential density (isopycnals). For shallow coastal regions, terrain-following coordinates may be more suitable to characterize the flow subject to the kinematic constraints provided by the bathymetry. In the surface mixed layer or where the ocean is un-stratified, fixed pressure level coordinates may better represent the flow. The detailed choices for vertical coordinates for HYCOM is discussed in Chassignet et al. (2003).

The primitive equations of the HYCOM are detailed in Bleck (2002):

$$\frac{\partial \boldsymbol{v}}{\partial t_s} + \nabla_s \frac{\boldsymbol{v}^2}{2} + (\zeta + f)\boldsymbol{k} \times \boldsymbol{v} + \left(\dot{s}\frac{\partial p}{\partial s}\right)\frac{\partial \boldsymbol{v}}{\partial p} \nabla_s M - p\nabla_s \alpha$$
$$= -g\frac{\partial \boldsymbol{\tau}}{\partial p} + \left(\frac{\partial p}{\partial s}\right)^2 \nabla_s \cdot \left(\frac{\partial p}{\partial s}\nabla_s \boldsymbol{v}\right) \tag{1}$$

$$\frac{\partial}{\partial t_s}\left(\frac{\partial p}{\partial s}\right) + \nabla_s \cdot \left(\boldsymbol{v}\frac{\partial p}{\partial s}\right) + \frac{\partial}{\partial s}\left(\dot{s}\frac{\partial p}{\partial s}\right) = 0 \tag{2}$$

$$\frac{\partial}{\partial t_s}\left(\frac{\partial p}{\partial s}\theta\right) + \nabla_s \cdot \left(\boldsymbol{v}\frac{\partial p}{\partial s}\theta\right) + \frac{\partial}{\partial s}\left(\dot{s}\frac{\partial p}{\partial s}\theta\right) = \nabla_s \cdot \left(\nu\frac{\partial p}{\partial s}\nabla_s \theta\right) + H_\theta \tag{3}$$

where $\boldsymbol{v}$ is the horizontal velocity vector, $s$ is the vertical coordinate, $\zeta$ is the relative vorticity, $f$ is the Coriolis parameter, $\boldsymbol{k}$ is the vertical unit vector, $p$ is pressure, $M = gz + p\alpha$ is the Montgomery potential, $\alpha$ is the potential specific volume, $\boldsymbol{\tau}$ is the horizontal wind stress at the surface or drag at the ocean bottom, $\theta$ are one of two thermodynamic variables, either temperature or salinity, and $\nu$ is the eddy viscosity coefficient. The first equation (1) is the momentum equation for the components of $\boldsymbol{v}$, yielding two scalar equations. The second equation (2) is the mass continuity equation. The third equation (3) represents two scalar thermodynamic equations, one for each thermodynamic variable. Thus, there are a total of five equations that are being solved. A unique feature of HYCOM is that the vertical coordinate system can be modified at any given time step during model integration as flow conditions change. This is done through the use of a grid generator.

In addition to the equations above, HYCOM includes parameterizations that take into account other physical processes, such as vertical mixing (possibly due to turbulence), convection and sea ice. The HYCOM model is a highly configurable model that can be run at a wide range of horizontal resolutions, vertical levels and can be driven using readily available lateral and boundary conditions (e.g., surface wind-forcing, tidal forcing and bathymetry).

## 2.1 HYCOM Data Assimilation System

The HYCOM modeling system in this study utilizes the T-SIS data assimilation system (Srinivasan et al., 2022). In the earliest version of this system, T-SIS followed the classical Kalman filter approach for optimal interpolation. In this approach, it is assumed that the model forecast follows a Markov process, which means that the future state of the system depends only on its current state and not on any previous states (Davis, 2013). Observations can improve the estimate of the model state in a least squares sense, taking into account the modeled and observed error covariances as follows:

$$\boldsymbol{x}_t^f = \boldsymbol{f}_t\left(\boldsymbol{x}_{t-1}^a\right) \tag{4}$$
$$\boldsymbol{x}_t^a = \boldsymbol{x}_t^f + \boldsymbol{K}_t\left(\boldsymbol{y}_t - \boldsymbol{H}_t\boldsymbol{x}_t^f\right) \tag{5}$$

where $\boldsymbol{x}$ is the model state, $\boldsymbol{f}$ refers to the forecast operator ("the ocean model"), while the $a$ superscript refers to the analysis after observations are assimilated. The matrix $\boldsymbol{K}$ is commonly known as the *Kalman Gain* matrix, it determines the relative weight given to the observations versus the forecast by taking into account model and observation error covariances. $\boldsymbol{H}$ is an observation operator that maps the modeled state variables to the observation variables. In the simplest scenario where the observations represent the same fields and have the same spatial and temporal resolution as the model, $\boldsymbol{H}$ is just the identity operator. In most cases, observations will sample only part of the model state, hence $\boldsymbol{H}$ will then interpolate the corresponding field(s) of the model state and perform any other transformation that may be needed. The Kalman gain is computed as

$$\boldsymbol{K}_t = \boldsymbol{P}_t^f \boldsymbol{H}_t^T \left(\boldsymbol{H}_t \boldsymbol{P}_t \boldsymbol{H}_t^T + \boldsymbol{R}_t\right)^{-1} \tag{6}$$

where $\boldsymbol{P}^f$ is the forecast model state error covariance matrix, and $\boldsymbol{R}$ is the observation error covariance matrix. When the observation errors are high ($\boldsymbol{R}$ is large), $\boldsymbol{K}$ gives low weight in the second term in Eq. 5, giving the forecast of the model more weight. In version 2.0 of T-SIS Srinivasan et al. (2022), an alternative approach to calculate the Kalman gain is used

$$\boldsymbol{K}_t = \left(\boldsymbol{P}_t^{-1} + \boldsymbol{H}_t^T \boldsymbol{R}_t^{-1} \boldsymbol{H}_t\right)^{-1} \boldsymbol{H}_t^T \boldsymbol{R}_t^{-1} \tag{7}$$

The Kalman gain is computed by first defining the information matrix as $\boldsymbol{L} \equiv \boldsymbol{P}^{-1}$ and then the information matrix is modeled by a Gaussian Markov Random Field (GMRF), which is a probabilistic model consisting of a set of random variables having a multivariate Gaussian distribution, with the Markov property that each variable is conditionally independent of all others given its immediate neighbors (Rue and Held, 2005). This property leads to a sparse precision (information) matrix $\boldsymbol{L}$, making computations more efficient. Each element is conditionally specified based on a set of neighbors. Via spatial regression (Chin et al., 1999), the neighbors can be determined in a manner that can lead to a sparse matrix for $\boldsymbol{L}$. This approximation of the inverse error covariance matrix results in a significant reduction in computational expense when used implicitly to solve Eq. (7). Speed-ups of an order of magnitude have been reported in Srinivasan et al. (2022).

Finally, after each assimilation step, there are further adjustments to the data in order to accommodate certain HYCOM constraints, such as model layer thickness adjustments, min/max thresholds, hydrostatic checks, and geostrophic balance. The data used in the assimilation have wide temporal and spatial availability.

## 2.2 Convolutional Neural Networks

Fully-connected neural networks, or dense networks, have approximately $m*n+n$ number of parameters for every layer whith $m$ previous nodes and $n$ current nodes. The number of parameters grows rapidly by incorporating additional intermediate layers, which are commonly needed to create complex models capable of approximating non-linear sytems. This makes dense networks impractical for training large-scale problems encountered in domains like computer vision, where each pixel in an image represent an input feature into the model. However, this limitation is overcomed by the introduction of Convolutional Neural Networks (CNNs).

CNNs are able to reduce the number of parameters in a neural network by sharing weights across different locations in the input data (LeCun et al., 1998). In CNNs, each neuron in a layer is connected only to a small region of the layer before it. This region is called the receptive field. This is from an inductive bias coming from the assumption that only nearby pixels in the images are likely to be related to each other, thus capturing local features in the input data like edges, textures, etc. CNNs are designed to process data with a grid-like topology (e.g. images).

Convolutional Neural Networks (CNNs) employ convolutions, a specialized type of linear operation, instead of general matrix multiplication in their layers (O'Shea and Nash, 2015). A convolution involves computing the output (feature map) by applying a filter (kernel) across the input, capturing local dependencies among input features. This operation is defined for sequences $a$ and $k$ as:

$$\boldsymbol{b} = (b_i), \quad b_i := \sum_{j=-n}^{m} a_j k_{i-j} \tag{8}$$

where $a$ is the input, $k$ the kernel, and $b$ the resulting feature map, demonstrating local connectivity. CNN architectures typically combine convolutional layers with pooling layers, which reduce the spatial dimensionality of feature maps, thereby decreasing the number of operations and enhancing computational efficiency (O'Shea and Nash, 2015).

U-nets, originally developed by Ronneberger et al. (2015) for biomedical image segmentation, are a type of Convolutional Neural Network (CNN) characterized by a symmetric encoder-decoder architecture forming a U-shape (Ronneberger et al., 2015). The encoder path, or contracting path, consists of repeated applications of convolutional layers followed by pooling layers, which progressively downsample the input. At each downsampling step, the number of channels is typically doubled, enabling the network to capture increasingly abstract features from the input data. The decoder path, upsamples the feature maps using transposed convolutions or other upsampling techniques. This path reduces the number of channels by half at each step and at the end reconstructs the spatial dimensions of the original input. A main feature of U-nets is the inclusion of skip connections between corresponding layers in the encoder and decoder paths. These connections concatenate feature maps from the encoder directly to the decoder, allowing the network to retain high-resolution features that might otherwise be lost during

downsampling. This design effectively preserves spatial information and enables precise localization, which is essential for tasks like segmentation.

Figure 1 illustrates a comparison between the classical Convolutional Neural Network (CNN) architecture (top panel) and the U-Net architecture (bottom panel). As previously described, U-Nets incorporate three additional components that distinguish them from traditional CNNs:

- Pooling Layers: Represented by red blocks in the figure, these layers progressively reduce the spatial dimensions of the feature maps.

- Upsampling Convolutions: Shown as green blocks, these layers increase the spatial resolution of the feature maps.

- Skip Connections: Depicted by blue arrows, these connections concatenate feature maps from corresponding layers in the encoder and decoder paths.

In this example, both architectures receive an input image of size $256 \times 256$. The classical CNN employs convolutional layers with a uniform configuration of 64 filters each. In contrast, the U-Net architecture uses a varying number of filters across its layers. Both architectures consist of ten convolutional layers, trying to maintain a comparable number of parameters and similar computational operations.

Over the past decade, U-Net architectures have been extensively applied to a variety of geoscience problems due to their capability to learn hierarchical features and capture both local and global contexts. Variants of U-Net, such as the attention U-Net (Oktay et al., 2018) and the Small Attention-UNet (SmaAt-UNet) (Roy et al., 2018), have been developed to enhance feature extraction and improve performance in complex geoscientific tasks. These variants introduce mechanisms like attention gates and efficient channel interdependencies, allowing models to focus on relevant features while reducing computational requirements. Some notable applications of the use of U-Nets in geoscience include:

- Remote Sensing and Earth Observation: U-Nets have been extensively used for semantic segmentation and classification of satellite imagery, including land cover mapping (Russwurm and Korner, 2018), building and road extraction (Maggiori et al., 2017; Demir et al., 2018), and change detection (Daudt et al., 2018);

- Meteorology and Climate Science: U-Net architectures have been employed for precipitation nowcasting using radar data (Agrawal et al., 2019);

- Hydrology and Flood Mapping: U-Nets have been applied to flood detection and mapping from satellite images (Pech-May et al., 2024), and mountain ice segmentation (Tian et al., 2022);

- Oceanography: U-Net architectures have been utilized in oceanography for bathymetry estimation from optical imagery (Nicolas et al., 2023), and ocean eddy detection and classification (Lguensat et al., 2018);

- Data Assimilation and Ocean Modeling: Beauchamp et al. (2022) introduced Multimodal 4DVarNets, where U-Net-based architectures obtain similar results to 4DVarNets for the reconstruction of sea surface dynamics by leveraging

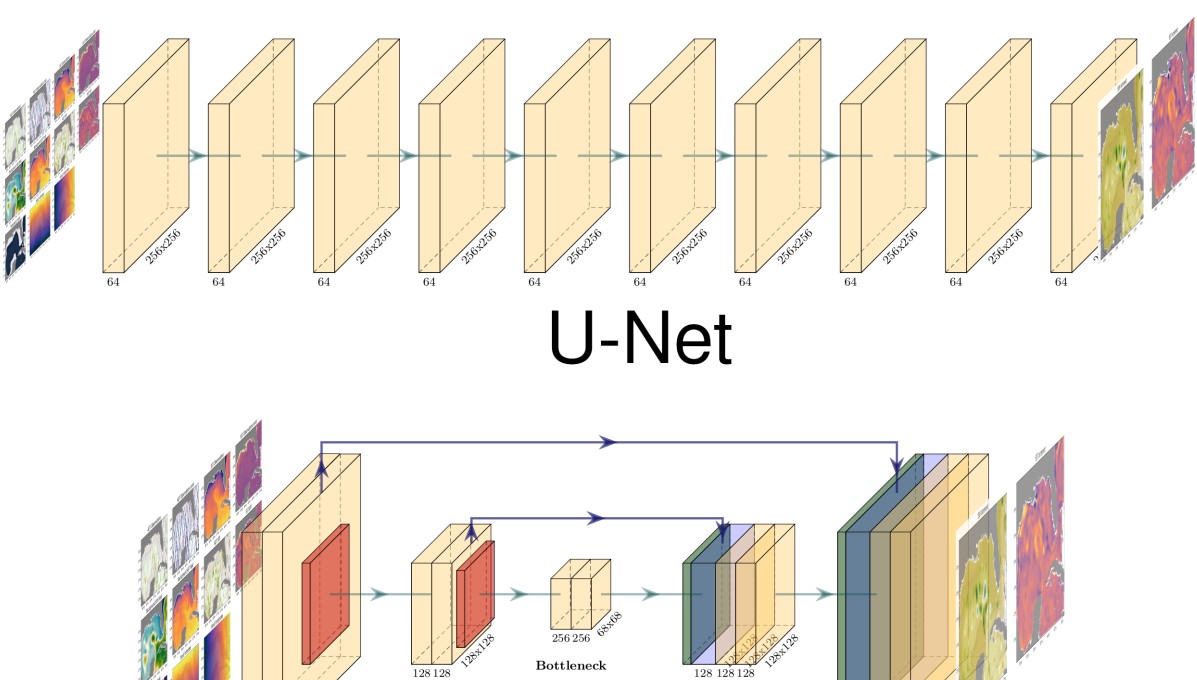

**Figure 1.** Comparison of the classical CNN architecture (top) vs the U-Net architecture (bottom).

synergies between sea surface temperature (SST) and sea surface height (SSH) observations. Their work demonstrates the capability of deep learning models to assimilate multiple data modalities and reconstruct ocean surface variables with high accuracy.

The versatility of U-Net architectures in geoscientific applications makes them a suitable choice for data assimilation in ocean modeling, given their ability to capture spatial dependencies and manage multiscale features. This capability aligns well with the demands of integrating observational data into ocean models, motivating our choice to adopt this architecture.

## 3   Data assimilation with Convolutional Neural Networks

In this section, we explore the use of Convolutional Neural Networks (CNNs) as a data assimilation technique for ocean
models. We assess the performance of multiple CNN models across five experimental setups. These models assimilate data from sea surface temperature (SST) and sea surface height (SSH) observations. Their performance is compared with results

obtained through the optimal interpolation method implemented in the T-SIS. The experiments are conducted in the Gulf of Mexico, covering a domain from $18.09°$ to $31.96°$ latitude and $-98.0°$ to $-77.04°$ longitude, as depicted in Figure 2.

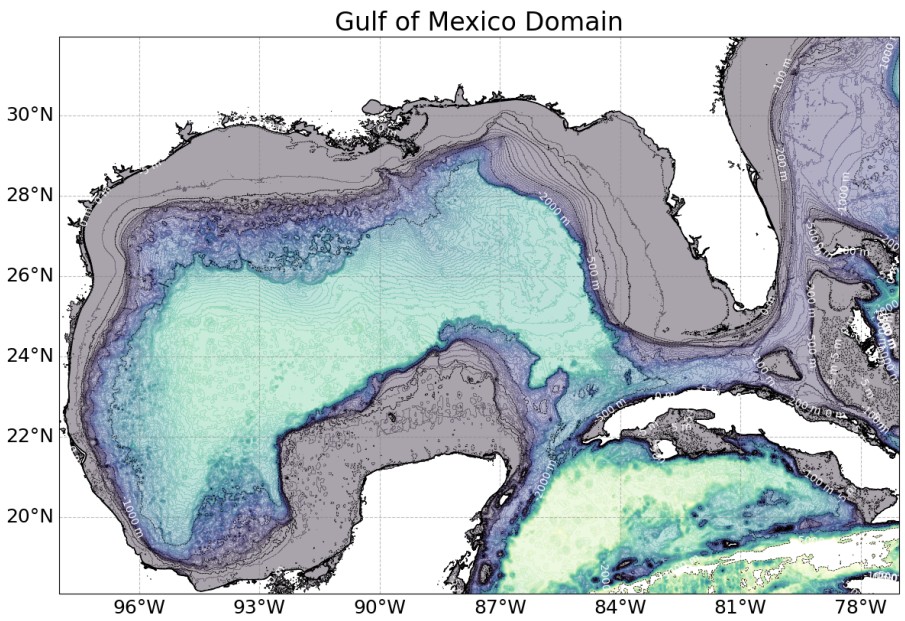

**Figure 2.** This map illustrates the geographic limits within the Gulf of Mexico used for our experiments.

## 3.1 Data

The ocean model used is the HYbrid Coordinate Ocean Model (HYCOM) with a spatial resolution of $1/25°$. The GOMb0.04 domain is set up with the high resolution 1km bathymetry of the Gulf of Mexico (Panagiotis, 2014) over a domain going from $98°E$ to $77°E$ in longitude and from $18°N$ to $32°N$ in latitude. With 41-hybrid layers in the vertical, the latest version of the HYCOM model (2.3.01: https://github.com/HYCOM/HYCOM-src) is forced at the surface with the CFSR/CFSv2 hourly atmospheric forcing. The lateral open boundaries are relaxed to daily means of the global HYCOM GOFS3.1 reanalysis

(https://www.hycom.org/dataserver/gofs-3pt1/reanalysis). The initial conditions are taken from a 20-year reanalysis created with the same configuration.

   The T-SIS package, detailed in section 2.1, is utilized with HYCOM for producing the *increments*, which are used to correct the model state and generate the *analysis*. To optimize the system's performance for the HYCOM Lagrangian vertical coordinate system, subsurface profile observations are first layerized (re-mapped onto the model's hybrid isopycnic-sigma-z

vertical coordinate system) prior to assimilation. The analysis procedure then updates each layer separately in a vertically decoupled manner. A layerized version of the Cooper and Haines (1996) procedure is used to adjust model layer thicknesses in the isopycnic-coordinate interior in response to SSH anomaly *innovations* (differences between observed values and the background state). In the data assimilation field, innovations refer to the differences between the observed values and the model's

background (prior) estimates of those values, expressed in the observation's frame of reference. Before calculating SSH inno-
vations the mean dynamic topography (MDT) is added to the altimetry observations. A MDT derived from a 20-year freerun
of the GOMb0.04 configuration is used for converting SLA to SSH. The multiscale sequential assimilation scheme based on
a simplified ensemble Kalman Filter (Evensen, 2003; Oke et al., 2002) is used to combine the observations and the model to
produce best estimates of the ocean state at analysis time.

To train the CNN models, we use real satellite observations of sea surface temperature (SST) from the Group for High Res-
olution Sea Surface Temperature (GHRSST) dataset and along-track altimeter sea surface height (SSH) observations (ADT).
The model background state, representing the previous forecast from HYCOM, is also used as input. Additionally, the *incre-
ments* are also used as input, which are the differences between the observations and the model background state. The CNNs are
trained to replicate the increments generated by the T-SIS data assimilation package. In summary, the CNNs learn to map the
background state, observations, and innovations to the increments, effectively emulating the data assimilation step performed
by T-SIS.

It's important to note that while the SST observations from GHRSST provide near-complete spatial coverage, the SSH
observations from along-track altimeter data are sparse and irregularly distributed. The DA schemes are able to handle such
sparse datasets and propagate the observational information across the model domain. This is achieved through statistical
interpolation and the physical dynamics represented in the model by the T-SIS system, which together allow us to estimate the
ocean state in unobserved areas based on the available observations.

This assimilative ocean model configuration is initially run for two years (2009 and 2010), generating a total of 730 daily
outputs. These outputs are used to train and validate the proposed CNN models. Each day's increment fields of SSH and SST,
$\boldsymbol{K}_t(\boldsymbol{y}_t - \boldsymbol{H}_t\boldsymbol{x}_t^f)$, the background state $\boldsymbol{x}_t^f$, and the observations $\boldsymbol{y}_t$ are employed to train the CNN models.

In Earth sciences, particularly in ocean modeling, data leakage is a significant concern due to the strong temporal autocorre-
lation in the data. The state of the ocean does not change dramatically over short periods, which means that random splitting of
data can lead to leakage where the model learns from future information. To mitigate this, we employed a chronological data
splitting strategy. From the 730 daily examples the first 80% is used for training, 10% for validation, and the last 10% is used
for testing, ranging from October 19[th] to December 31[st] of 2010. This method ensures that the model is trained on past data
and evaluated on future data, reducing the risk of information from the test set influencing the training process. However, we
recognize that the proximity of the training and test sets may still allow for some data leakage due to the ocean's slow-changing
nature.

To further assess the model's ability to generalize and to address potential data leakage, we tested the model on datasets from
the years 2002 and 2006. These years were selected because they exhibit different dynamical states of the GoM, with the Loop
Current mostly in retracted and extended phases, respectively. By evaluating the model on data that is entirely separate from
the training and validation sets and representing different oceanographic conditions, we reduce the likelihood that the model's
performance is artificially inflated due to data leakage.

The model maintained strong performance on these additional datasets, with RMSE values comparable to those on the
original test set as described in the Generalization Tests section.

## 3.2 Preprocessing

Prior to training the Convolutional Neural Network (CNN) models, we performed several preprocessing steps to ensure that the input data was appropriately scaled and formatted. First, to address the issue of differing value ranges among the input variables, we normalized each field individually. This normalization involved adjusting each input field—such as sea surface temperature (SST) observations and sea surface height (SSH) observations—to have a mean of zero and a standard deviation of one. Normalization is an important step in machine learning, as it ensures that all input features contribute equally during

training, preventing variables with larger magnitudes from disproportionately influencing the model's learning process. By standardizing the inputs, we facilitated a more stable and efficient optimization during model training.

The parameters used for normalization, specifically the mean and standard deviation for each input field, were calculated using the data from the full training period, encompassing the years 2009 and 2010. These calculated parameters were then applied to the validation and test datasets, as well as to the additionally tested years 2002 and 2006.

After the CNN models generated the predicted increments, we applied an inverse transformation using the previously calculated mean and standard deviation to denormalize the outputs. This denormalization step converted the increments back to their original units—such as degrees Celsius for SST or meters for SSH—making them compatible with the model forecast corrections. By restoring the original scale of the data, we ensured that the increments could be directly applied to the HYCOM model outputs.

We addressed the irregular distribution and missing values of the along-track altimeter SSH data by mapping these observations onto the model grid and filling the missing data points with zeros. Representing the absence of observations with zeros allowed the CNN models to process the SSH data as continuous fields, where zeros explicitly indicated locations without observational data. The response of the CNN to missing values represented as zeros is of interest to us and was part of the experiments.

The loss function was computed only over ocean grid points. This ensured that the network focused solely on learning the ocean dynamics and was not penalized for predictions over land. After the models are trained and used for predictions, we applied the land mask to set the values at land grid points to NaN. This step ensured that the output fields contained valid data only over ocean areas, aligning with the physical reality that oceanographic variables are undefined over land.

Additionally, since T-SIS does not provide SSH increments for shallow areas (depths less than 200 meters), we included a

265 mask indicating areas with depths greater than 200 meters. This depth mask was provided as an additional input channel to help the CNN learn this restriction and avoid predicting increments in shallow regions where T-SIS does not apply corrections.

The input tensors to the CNN models are four-dimensional arrays with the following dimensions: [Batch Size, Height, Width, Channels]. The numbers for Height, Width, and Channels vary depending on the experiment. All input tensors are of type float32.

## 3.3 Experiments

The CNNs' performance is assessed through five controlled experiments designed to test the expected behavior in practical operational settings with full primitive equations, real observations, and complex topographies. These experiments investigate the CNNs' response relative to the size of the spatial windows used for model training, the complexity of the CNN architecture, the number and types of ocean fields used as input and output fields, and the allowed ocean percentage in the training examples. In the experiment analyzing network complexity, we evaluated different network complexities by comparing Simple CNN architectures with varying depths (2, 4, 8, and 16 layers) to the U-Net architecture. This experiment aims to assess the impact of network depth on model performance. All subsequent experiments utilize the U-Net architecture exclusively to explore the effects of window size, input configurations, ocean percentage, etc.

It is important to note that our experiments are not twin experiments. In twin experiments, synthetic observations are generated from a model run (considered the "truth") and are then assimilated back into the model to assess the data assimilation system under controlled conditions. In our study, real observational data is used for both training and testing our CNN models. The T-SIS data assimilation system generates increments based on these real observations, and our CNN models are trained to replicate these increments. By using actual observations from GHRSST for SST and along-track altimeter data for SSH, our experiments reflect a more realistic scenario where the CNN models learn from real-world data, capturing the complexities and uncertainties inherent in operational ocean modeling.

Figure 3 shows an example of all the possible inputs and outputs tested in the experiments. The first row of the inputs display the sparse SSH observations and its associated error. The second row displays the model background state and the difference with respect to the observation. The third row shows additional inputs for the CNN that can improve the prediction (200m mask and normalized latitude and longitudes). The final row show the SSH and SST increments produced by T-SIS and learned by the CNN model. Despite the sparsity of the SSH observations, the CNN models produce complete increment fields by propagating the information throughout the domain.

### 3.3.1 Window size

The first experiment examines the CNNs' performance relative to the size of spatial windows used as input. Training a CNN within a fixed domain does not guarantee effective generalization to other domains. Despite the translational invariance of convolutional layers, models may develop biases based on the specific features, such as land-sea boundaries, within the training domain, making it challenging to generalize to domains with different coastlines and ocean dynamics. Conversely, using the entire domain as the training set provides only one training example per day. Training with smaller windows increases the number of examples, potentially enhancing generalization but possibly at the expense of losing context provided by the larger domain.

Training a model with the full domain provides just one training example per day. When training with smaller window sizes, the total number of training examples is determined by how many sub-windows can fit within our domain. For each dimension,

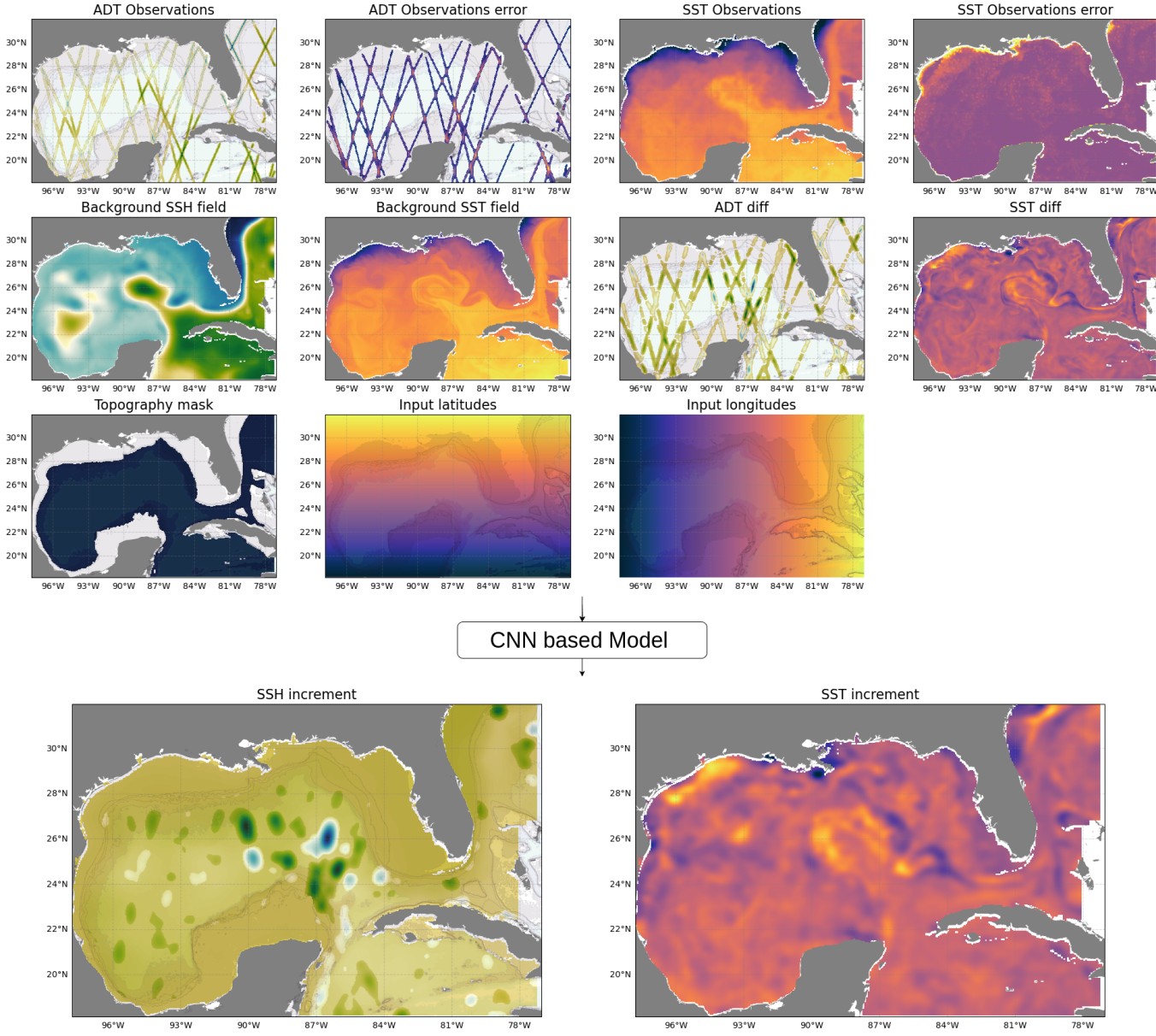

**Figure 3.** Example of all the possible inputs and outputs tested in the experiments. The first row of the inputs display the sparse SSH observations and its associated error. The second row displays the model background state and the difference with respect to the observation. The third row shows additional inputs for the CNN that can improve the prediction (200m mask and normalized latitude and longitudes). The final row show the SSH and SST increments produced by T-SIS and learned by the CNN model. Despite the sparsity of the SSH observations, the CNN models produce complete increment fields by propagating the information throughout the domain.

the total amount of sub-windows that can be selected is given by:

$$\text{Number of training examples} = D_s - W_s + 1 \tag{9}$$

Here, $D_s$ represents the dimension size, and $W_s$ denotes the window size. For instance, if the domain size is $10 \times 10$ and the window size is $5 \times 5$, we can fit a total of 6 windows in each dimension, or a total of 36 different examples. In our experiments, when training the networks with a window size smaller than the full domain, 10 random windows are selected for a given day, and each epoch is completed after 1000 of these randomly selected windows are generated. The random images change betwen batches and epochs. The experiment compares performance across four window sizes: the entire domain ($384 \times 520$ pixels), and smaller windows of $160 \times 160$, $120 \times 120$, and $80 \times 80$ pixels.

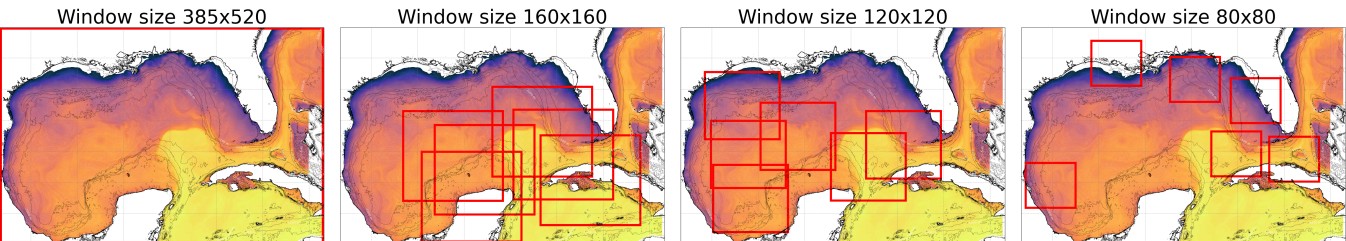

**Figure 4.** Examples of randomly selected training windows at various sizes: 385x520 pixels (full domain), 160x160, 120x120, and 80x80 pixels

To assess the impact of spatial coordinates on model performance, we conducted additional experiments by including normalized latitude and longitude fields as input channels to the network. The latitude and longitude values were scaled between 0 and 1 to align with the normalization of other input features.

### 3.3.2 CNN complexity

The second experiment evaluates the performance of the CNNs for data assimilation concerning the complexity of the CNN architecture. Five different models are evaluated using two CNN architectures. The first four models follow a simple CNN architecture, which we refer to as *SimpleCNN*. Models from this architecture are built by stacking convolutional layers with an increasing number of filters. Each hidden convolutional layer employs a ReLu activation function, and the last two convolutional layers contain a single filter and a linear activation function. The four models using this architecture vary in the number of hidden convolutional layers with 2, 4, 8, and 16 layers. The second architecture tested follows the encoder-decoder architecture with skip connections from the U-Net (Ronneberger et al., 2015). For this architecture, one model with three levels and 18 CNN layers is evaluated. Each convolutional layer in the U-Net architecture is followed by a ReLu activation function and a Batch Normalization layer, except for the final output layer. The inclusion of Batch Normalization helps stabilize and accelerate training and provides regularization benefits by reducing internal covariate shift. Figure 5 presents detailed information on this model, where all CNN layers except the last one use the ReLu activation function.

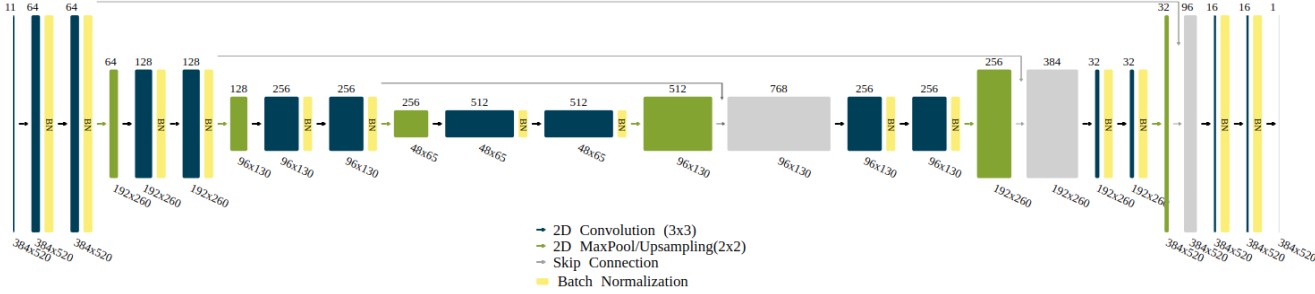

**Figure 5.** Detailed illustration of the U-Net architecture employed in the experiments.

Table 1 shows the names, number of hidden layers, and number of filters used at each hidden layer for the five model architectures tested.

**Table 1.** Summary of the number of hidden layers and filters tested in each of the proposed CNN architectures.

| Name | CNN Hidden layers | Filter size |
|------|------|------|
| SimpleCNN02 | 2 | 32, 64 |
| SimpleCNN04 | 4 | 32, 64x3 |
| SimpleCNN08 | 8 | 32, 64x7 |
| SimpleCNN16 | 16 | 32, 64x15 |
| U-Net | 14 | 16x2, 32x2, 64x2, 128x2, 64x2, 32x2, 16x2 |

### 3.3.3 Input types

This experiment investigates the use of multiple ocean fields as inputs in our models. Traditional methods, which approximate the model's error covariance matrix, face scalability challenges when multiple fields are integrated into the assimilation process, significantly increasing the matrix size. By varying the input types, including SST, SSH, and their respective observation errors we evaluate the potential of a unified deep learning model to assimilate diverse data sources effectively.

### 3.3.4 Outputs

This experiment evaluates the network's performance concerning the type and number of output fields. As in the previous experiment, having a single model that can assimilate observations into multiple fields is desired. The two fields considered in this experiment are SSH and SST, tested individually and jointly. The primary objective is to investigate whether a moderately complex CNN model can assimilate observations from multiple fields as effectively as from a single field.

### 3.3.5 Percentage of Ocean

Given that CNNs were originally designed for image processing, they typically do not account for non-valid pixels like land areas. This experiment varies the minimum ocean area required in the training windows, testing thresholds of 0%, 30%, 60%, and 90%. For the 0% scenario, there are no restrictions imposed on the ocean coverage within the training windows, meaning these windows could entirely encompass land areas. Conversely, in the 90% scenario, any training windows containing less than 90% ocean coverage are excluded. The window size for this experiment is fixed at 160x160 pixels.

Table 2 summarizes the parameters and their respective values tested in our experiments. Each parameter was varied independently to assess its impact on the performance of the CNN models in assimilating oceanographic data. By exploring different combinations of window sizes, CNN complexities, ocean percentages, inputs, and outputs, we aimed to gain comprehensive insights into the behavior and capabilities of the models under various conditions. Each tested model is trained five times to gather statistics on the training's consistency and allow a more accurate comparison between the models' performances. A total of 75 CNN models are evaluated in these experiments.

**Table 2.** Parameters and their tested values in the experiments.

| Parameter | Values Tested |
|---|---|
| Window Size | $384 \times 520$, $160 \times 160$, $120 \times 120$, $80 \times 80$ |
| CNN Complexity | SimpleCNN_02, SimpleCNN_04, SimpleCNN_08, SimpleCNN_16, U-Net |
| Ocean Percentage | 0%, 30%, 60%, 90% |
| Inputs | SSH — SSH and SST — SSH, SST, SSH Error, and SST Error |
| Outputs | SSH — SST — SSH and SST |

### 3.4 Training hyperparameters

All models are trained using the Adam optimizer (Kingma and Ba, 2014) with a learning rate of $10^{-3}$. The loss function used is the Mean Square Error (MSE), evaluated between the increment provided by the CNN and the one generated by the T-SIS model. The MSE loss is only evaluated in the grid cells where there is ocean, and the CNN models' outputs are always masked by land areas, which are irrelevant for data assimilation in the ocean. We used a batch size of 32 to train the models. All trainings are ended when the error in the loss function of the validation set has not decreased for 20 epochs, the model with the lowest validation loss is used for the statistics.

We conducted initial experiments incorporating a Dropout rate of $20\%$ after the convolutional layers. However, we observed that the networks with Dropout exhibited lower performance compared to those without Dropout. This could be attributed to the size of our training dataset, which may not be large enough for Dropout to be effective. Consequently, we opted not to include Dropout in our final models to maintain optimal performance.

Regarding optimizer selection, we used the standard Adam optimizer with default parameters in our experiments. We acknowledge that the AdamW optimizer, which includes decoupled weight decay for L2 regularization, could potentially enhance

generalization by applying a stronger weight penalization. We plan to explore the use of AdamW with increased weight decay in future work to assess its impact on model performance.

## 4 Results

In this section we describe and analize the results from the proposed experiments to use CNN for data assimilation in ocean models. For each combination of parameters five models are trained, the error bar plots show the mean (orange line), median (green triangle) and standard deviation of the models. The statistics are obtained from the test set, with dates from October 19th to December 31st of 2010 . For all the experiments the y axis is the RMSE in meters (already denormalized) of the difference between the increment provided by T-SIS and the one provided by the CNN.

### 4.1 Window Size

Figure 6 shows a performance comparison with respect to the window size used to train the networks. In this experiment, all other parameters remain fixed, with U-Net serving as the default architecture. The SSH increment is used as the target output, and the SSH background state $x_t^f$ and satellite altimeter observations $y_t$ are used as inputs. Furthermore, a mask delimiting areas in the GoM deeper than 200 meters is included as input because T-SIS does not generate any SSH increment for shallow areas. To enable the CNN to learn this restriction, we provided this mask as an additional input channel.

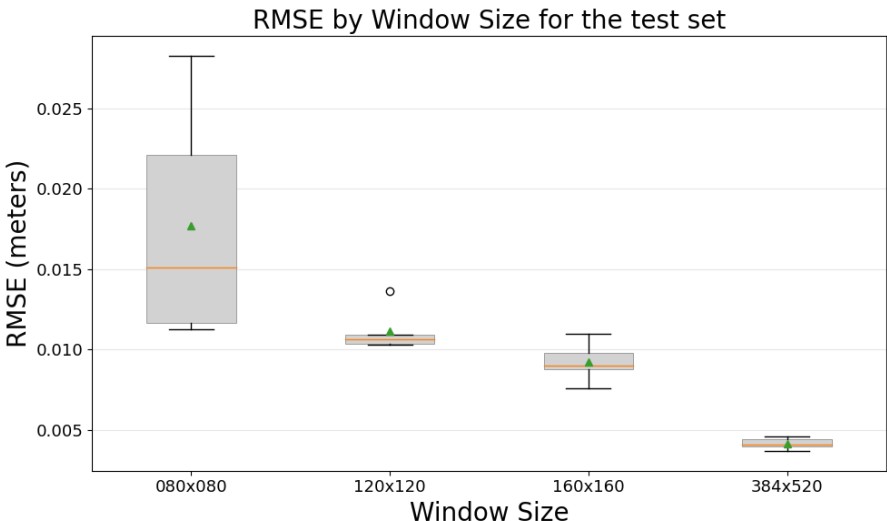

**Figure 6.** RMSE comparison between CNN models and the T-SIS method across different window sizes on the test dataset.

The experiment reveals a clear relationship between the model's performance and the size of the window used for training. Larger windows yield better performance, and using the entire domain for training achieves the best results. These results

indicate that the CNN is benefiting from the context of the full domain and is not being affected by the reduced number of training examples that this configuration generates.

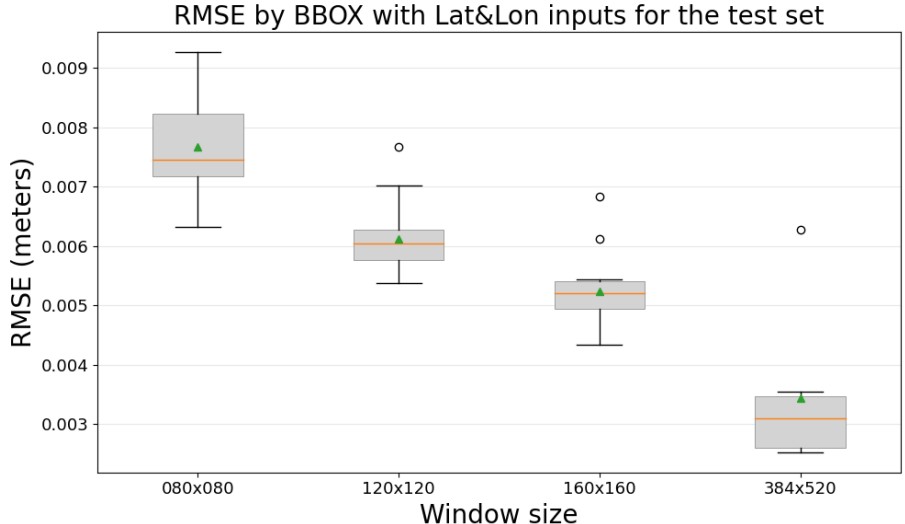

**Figure 7.** RMSE comparison between CNN models and the T-SIS method across different window sizes on the test dataset. With the inclusion of latitude and longitude as additional inputs.

The inclusion of latitude and longitude as additional inputs does reduce the RMSE of the models, specially those that are trained with smaller windows as shown in Figure 7. This indicates that the additional input layers provide useful spatial information to the models. The tendency still remains the same, larger windows yield better performance and the best results are obtained when the entire domain is used for training.

## 4.2   CNN complexity

Figure 8 presents the results of the comparison of the CNN architecture and complexity. As before, all other parameters remain fixed. In this case, we used the full domain to train the models, with SSH increment used as the target output, and SSH background state, shallow water mask, and satellite altimeter observations serving as input.

    The results illustrate that, for the problem of data assimilation in ocean models mimicking the optimal interpolation method, the CNNs' performance improves with increased complexity in their architecture. Two key observations include: the expo-
nential decay observed in the RMSE of the loss function relative to the complexity for the *SimpleCNN* architectures (as the number of hidden layers increases), and how the more advanced U-Net architecture, incorporating batch normalization, skip connections, and an encoder-decoder design, yields the best performance. It's worth noting from this experiment that although there's a clear relationship between the complexity of the CNNs' architectures and the performance obtained, the difference between them is not too big. The *SimpleCNN* architecture with only four hidden CNN layers already approximates the T-SIS
data assimilation package with a RMSE of just 8 mm.

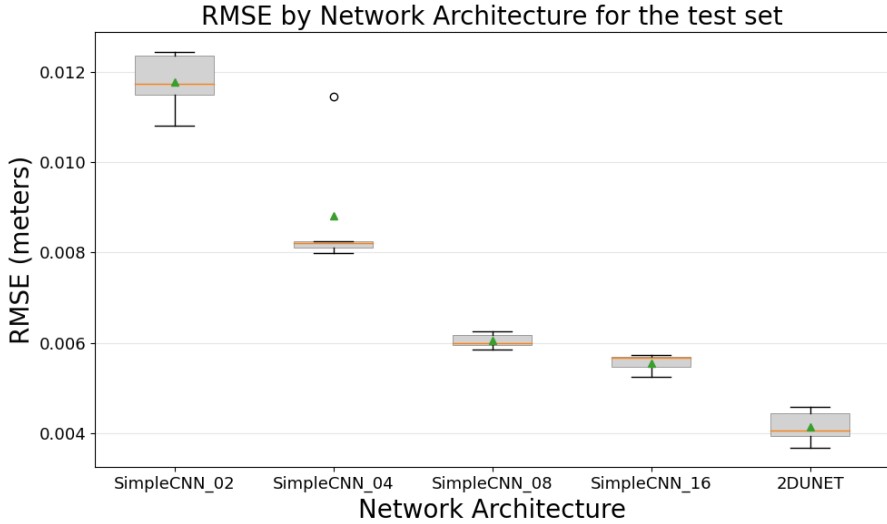

**Figure 8.** RMSE comparison between CNN models and the T-SIS method across different architectures on the test dataset.

### 4.3 Ocean Percentage

Figure 9 presents the results of the experiment that compares the percentage of ocean required in the training windows. Recall that the goal of this experiment is to investigate how grid cells with land areas can affect the training of the CNNs—a problem that is not common in computer vision problems. For this experiment, the window size is fixed at $160 \times 160$, the network architecture is the U-Net, the SSH increment is used as the target output, and the SSH background state and satellite altimeter observations are used as inputs.

Interestingly, we do not identify a clear trend between the performance of the CNNs and the percentage of ocean specified in the training examples. These results suggest that CNNs are not significantly affected by land grid cells when addressing the problem of data assimilation in ocean models.

### 4.4 Inputs

Figure 10 presents the results of including additional observations as input into the models. For this experiment, the rest of the parameters are as follows: U-Net is used as the network architecture, the entire domain is used to train the models, and the SSH increment serves as the target output. The three tested input observations are the satellite altimeter tracks (SSH), the altimeter tracks combined with SSH and their corresponding observational errors (SSH, SSH-ERR, SST, SST-ERR), and the altimeter tracks combined with SST, but without the error information (SSH, SST).

This experiment reveals how the CNN models might benefit from additional observations as inputs. The performance improves when the error of the observations is included as an input (as an extra channel in the input layer), and the variance of the trained models improves when including SST observation as an input variable. It is expected that the performance im-

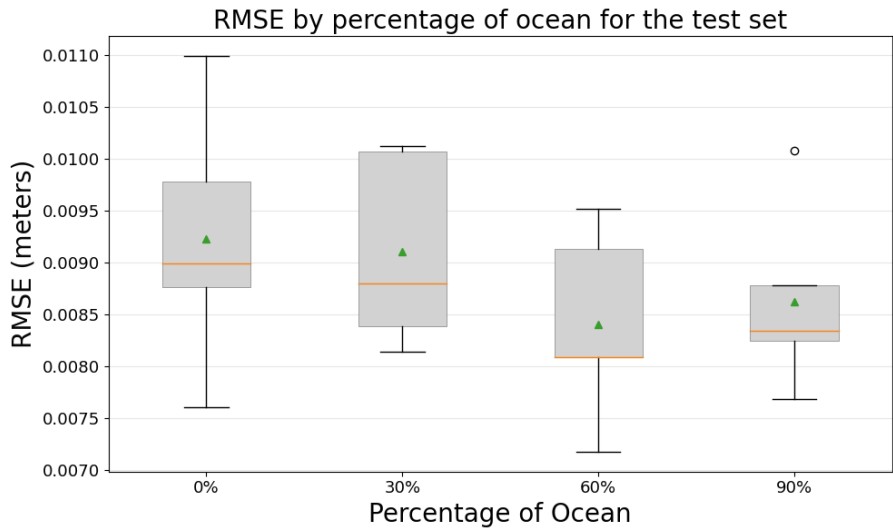

**Figure 9.** RMSE comparison between CNN models and the T-SIS method across different percentages of ocean areas in training examples on the test dataset.

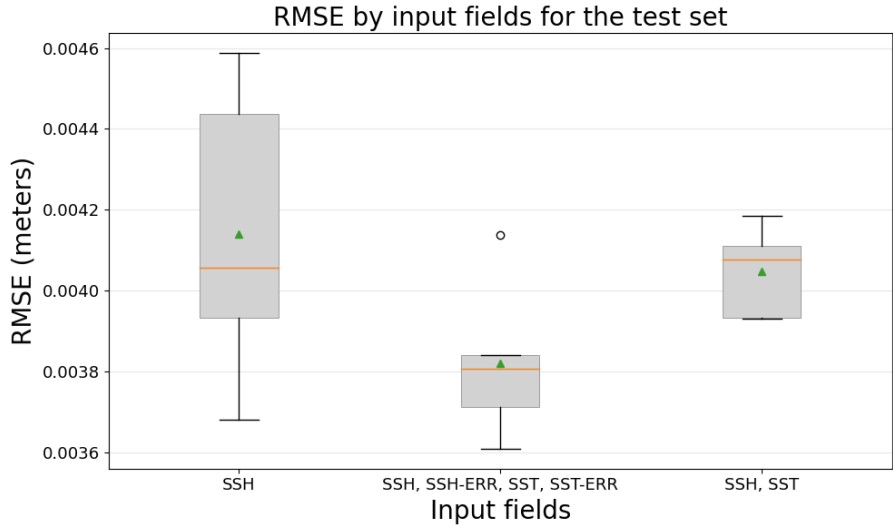

**Figure 10.** Comparison of the test RMSE loss by the number and types of input fields.

proves by including the observational error because T-SIS uses it to compute the increment—the error covariance matrix of the observations, $R_t^{-1}$ in equation 6, contains this information. However, it's noteworthy to show that including additional SST observations does not affect the model's performance, even though we know that SST is not used by T-SIS to generate the SSH increment.

## 4.5  Outputs

Finally, figure 11 presents the results of testing CNNs to simultaneously generate multiple data assimilation increments, in this case, SSH and SST. The rest of the parameters are as follows: U-Net is used as the network architecture, the entire domain is used to train the models, and the SSH increment serves as the target output. The three output increments tested are the satellite altimeter tracks (SSH), sea surface temperature (SST), and both together (SSH, SST).

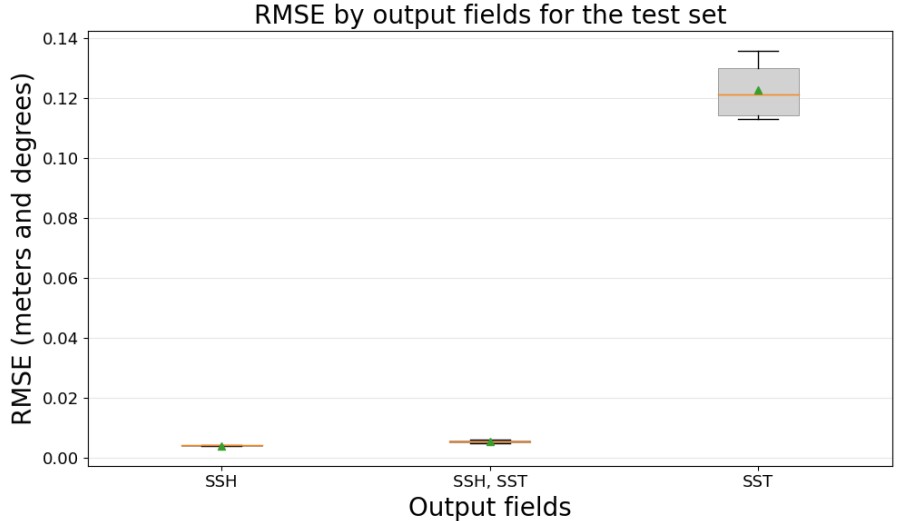

**Figure 11.** RMSE comparison of CNN models by the number and types of output fields, evaluated in the test dataset.

For this final experiment, it's important to note that the Y-axis is in meters for the first two models, SSH and SSH, SST, but it is in degrees for the last case of SST. The key takeaway from this experiment is that the performance in predicting the SSH increment is not affected when the model is tasked with generating both increments (SSH and SST) simultaneously. This indicates the ability of the CNNs to manage multiple outputs without a significant drop in performance for individual tasks.

## 4.6  Generalization Tests

Following the series of experiments in the previous section, which provide insights in the performance of CNNs in an operational ocean model setting with data assimilation, the best model was selected based on optimal parameters. This model utilized the U-Net architecture, was trained using the entire domain of the Gulf of Mexico (GoM) for training examples, and incorporated the SSH observations, the SSH observation errors, the SSH background state, and a binary mask indicating depths greater than 200 meters as inputs. The desired output was the increment of SSH, essentially the corrections to be made to this field in the model on a daily basis.

The ability to generate complete assimilated fields from sparse observations is a fundamental aspect of the data assimilation process. By taking advantage of the spatial covariance structures and dynamical relationships within the ocean model, the

DA scheme can infer the state of the ocean in regions without direct observations. Our CNN models are trained to learn this mapping from inputs (sparse observations and background state) to outputs (complete increments), effectively capturing the essence of the DA process eventhough the sparse observations are filled with zeros.

Figure 12 illustrates a comparison between the SSH increment as predicted by T-SIS and the increment predicted by the CNN model for a specific day, October 27th, 2010, from the test dataset. Generally, the overall predictions are similar, with the RMSE across the entire domain in this example being 3.2 mm. This day was randomly selected to provide a representative example of the model's performance in a typical scenario.

However, the figure also reveals some discrepancies, primarily at the peripheries of areas where there is an increment. This could potentially be attributed to a hard threshold within T-SIS that doesn't provide any increments beyond a certain distance from the observation. An expected pattern from Figure 12 is that the corrections to the model are made predominantly close to the locations of the satellite tracks.

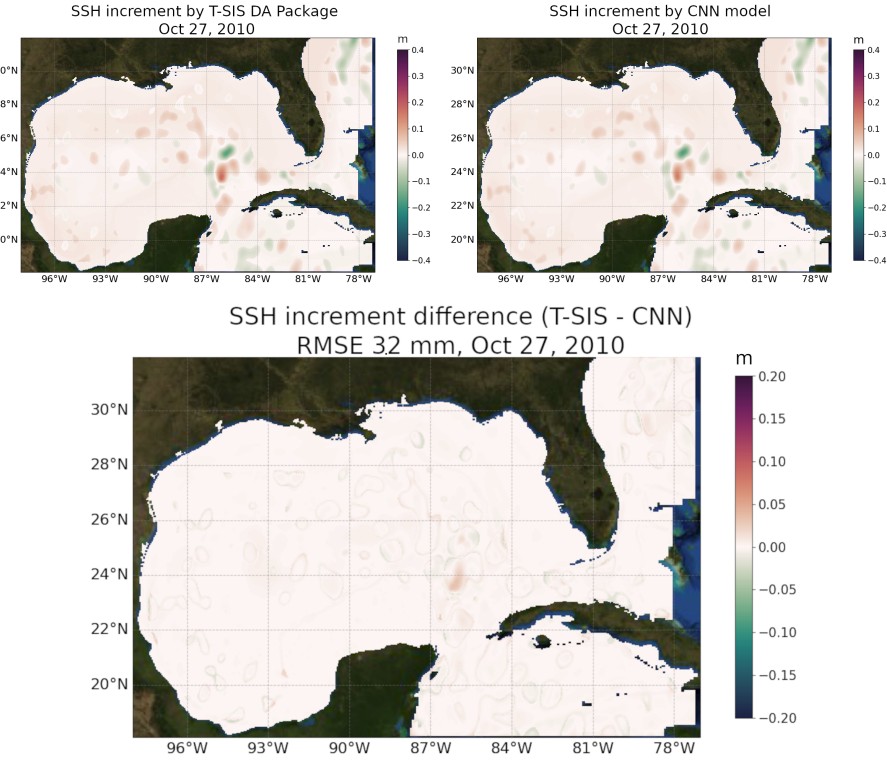

**Figure 12.** Comparison of predicted model error (increment) for sea surface height from the T-SIS in the top left panel, and from the proposed CNN model in the top right panel. The middle bottom panel displays the root mean square (RMS) error between both predictions.

Figure 13 depicts RMSE for the entire test set, ranging from October 19th to December 31st of 2010, as well as the initial days of the year used for training. The mean RMSE for all the test set is 3.72 mm, while for the days used for training it is

3.51 mm. This suggests that the CNN model is effectively generalizing to unseen examples. However, two points need to be considered in this analysis:

1. The Gulf of Mexico's dynamics do not change rapidly over time. Hence, the dynamical state of the GoM for the test set might be quite similar to the state used for training the model.

2. There is a slight discrepancy in the mean RMSE between the training and test sets, indicating that a more comprehensive experiment is necessary to understand how effectively the model generalizes to unseen examples where the GoM's dynamical state differs from that in the training set.

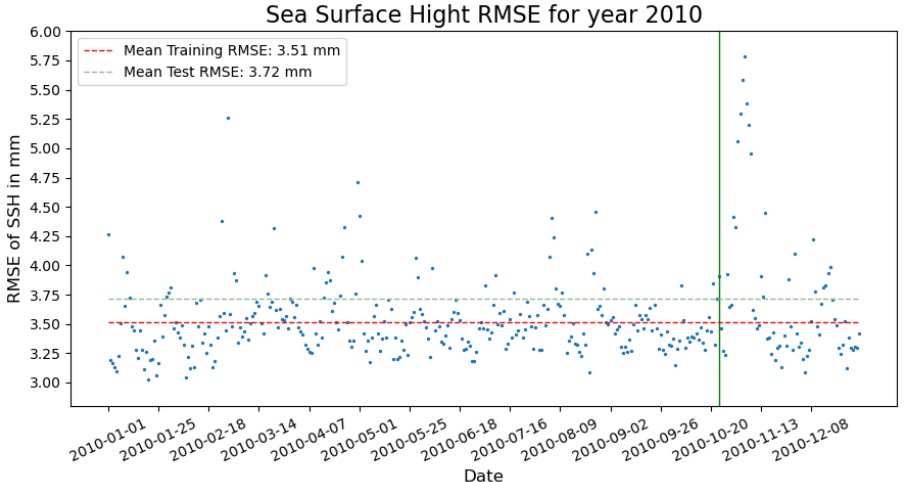

**Figure 13.** RMSE of the proposed CNN model for the year 2010. The vertical green line indicates the date where the test dataset starts. The two dashed lines indicate the RMSE of the training and test sets.

To scrutinize the model's ability to generalize across different dynamical states of the GoM, two contrasting years were chosen based on the states of the Loop Current (LC), the key driver of ocean dynamics in the GoM. Notably, it's challenging to confidently predict how a trained model will perform on unseen data. In experiments that use synthetic data, it is simpler to identify examples that fall outside of the training distribution, but in this scenario, the process is not as straightforward. The assumption is that the CNN model will learn to assimilate observations in the GoM comparable to the optimal interpolation method in T-SIS and will generalize correctly to data from different years, regardless of the GoM's dynamical state.

The years 2002 and 2006 were selected for this test. In 2002, the LC is primarily in a contracted state, while in 2006, it is predominantly in an extended state, with some eddies being shed throughout the year. New assimilated runs of HYCOM and T-SIS were created for these two years as described earlier, featuring a $1/25°$ spatial resolution and using NCEP CFSR/CFSv2 as the atmospheric forcings.

Figure 14 showcases a day from 2002 and 2006, emphasizing the different dynamical states of the GoM for these two years.

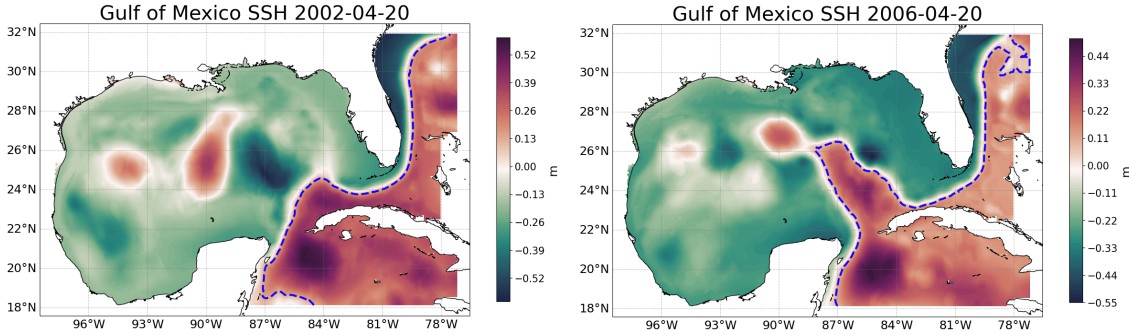

**Figure 14.** Contrasting dynamical states of the Gulf of Mexico for years 2002 and 2006. The left panel illustrates the retracted Loop Current on April 20[th], 2002, while the right panel depicts the extended Loop Current on April 20[th], 2006. Both cases are representative of the mean dynamical state of the GoM for that respective year.

The RMSE of the proposed model, trained with data from 2009 and 2010, is 4.39 mm for 2002 and 4.22 mm for 2006. This demonstrates how effectively the model is generalizing to new data and varying states of the GoM. It is anticipated that the model will yield similar results, with an RMSE around 4 mm, for any other timeframe of the GoM. Figure 15 presents the RMSE obtained for every day in 2002 and 2006, along with the mean for the two years. The RMSE has increased from 3.7 mm in the test set to 4.2 mm in this new generalization test. This underscores the importance of identifying appropriate scenarios to test the generalization of our models. Specifically, in the context of ocean models, it is crucial to evaluate the model in different dynamical scenarios than the ones used for training the models to avoid overestimating metrics that may not hold up when using the model operationally.

## 4.7 Performance Comparison

The primary objective of this study is to explore the use of Convolutional Neural Networks (CNNs) as a more efficient alternative to traditional data assimilation methods in oceanographic modeling. Comparing the performance between the proposed CNN model and the traditional T-SIS optimal interpolation method presents several challenges.

The proposed CNN model, still in the prototype stage, assimilates surface data for a single field at a time, and in some experiments, two fields. In contrast, the T-SIS data assimilation software is a fully operational package that simultaneously assimilates all HYCOM fields, including temperature, sea surface height, velocity fields U and V, salinity, in the 41 vertical layers of the model. Furthermore, the T-SIS package is implemented in FORTRAN and is typically run on clusters of tens to hundreds of CPUs at High Performance Computing (HPC) centers. Meanwhile, the proposed CNN model is implemented in TensorFlow with Python and utilizes GPUs, which may contain thousands of smaller processors.

In this performance analysis, we compare the times taken by T-SIS to assimilate a single day of observations under two different settings. The first setting involves execution on a cluster with 32 processors at the Florida State University HPC

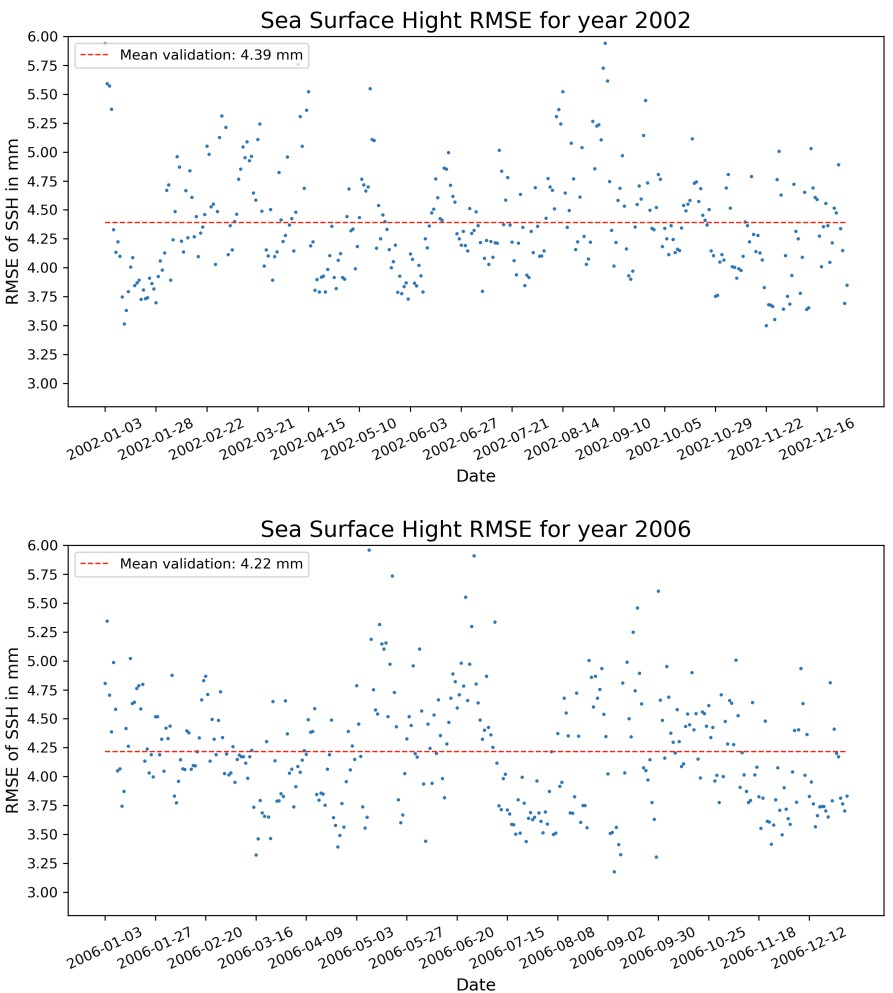

**Figure 15.** Root Mean Square Error (RMSE) of the proposed CNN model for the years 2002 (top panel) and 2006 (bottom panel). Vertical dashed lines represent the mean error across each respective year.

center, and the second on the Narwhal Navy Super Computer with 96 processors. The proposed CNN model, assimilating a single surface field (one day) takes $0.054 \pm 0.005$ seconds on an NVIDIA Quadro RTX 4500 GPU.

To estimate performance in a full 3D context (assimilating 5 fields across 41 vertical layers), we consider two scenarios. The first scenario, labeled *CNN Sequential*, assumes no further parallelization, requiring multiplication of our observed times by both the number of fields and vertical levels (41*5). The second scenario, *CNN Parallel*, assumes complete parallelization in 3D. The potential speedup of the proposed CNN model, compared to the 32-processor T-SIS, ranges from 1.9 to 389. Against the 96-processor T-SIS configuration, the speedup ranges from 0.73 (slower) to 150. The **speedup** or speedup factor is a unitless measure defined as the ratio of the time taken by the traditional T-SIS method to the time taken by our proposed CNN model

490

for the same assimilation task. For example, a speed-up factor of 58 indicates that the CNN model performs the assimilation 58 times faster than the T-SIS method running on a 32-processor cluster, while a speed-up factor of 22 signifies a 22-fold increase in performance compared to the 96-processor T-SIS configuration.

Given the broad range of these results, we simulate the assimilation of all vertical layers by running our model in batches of 41, providing a more practical metric. In this scenario *CNN 41 Batch*, it takes $0.36 \pm 0.01$ seconds to perform a single day's assimilation, resulting in more realistic expected speed-ups of 58 compared to the 32-processor T-SIS and 22 compared to the 96-processor T-SIS.

Table 3 displays the times, in seconds, taken by the T-SIS package to assimilate one day of data on an HPC with 32 and 96 processors, along with estimates for times our model would require, assuming the ability to parallelize only the vertical layers as a batch and when full parallelization of the five fields is possible.

**Table 3.** Comparison of simulation times for a single day of assimilated observations using T-SIS across various CPU configurations and the proposed CNN model with multiple estimated parallelization schemes.

|  | T-SIS 32 Procs | T-SIS 96 Procs | CNN Sequential | CNN Parallel | CNN 41 Batch |
|---|---|---|---|---|---|
| **Time** | $21 \pm 0.3$ s | $8.11 \pm 0.5$ s | $11.07 \pm 0.1$ s | $0.054 \pm 0.005$ s | $\mathbf{0.36 \pm 0.01}$ s |
| **Speedup vs 32 Proc** | 1x | 2.59x | 1.90x | 388.89x | **58.33**x |
| **Speedup vs 96 Proc** | 0.38x | 1x | 0.73x | 150.19x | **22.53**x |

# 5 Conclusions

In this work, we conducted a series of experiments to analyze the performance of Convolutional Neural Networks (CNNs) in emulating the data assimilation process within a realistic operational model setting. These experiments assessed various aspects of the CNNs, including architecture complexity, the types and quantities of observations (inputs), assimilated fields (outputs), responses to window size, and the influence of coastline on model performance.

The results demonstrated a clear relationship between the training window size and performance; larger window sizes generally better results, particularly when the full domain was used as the training window. Our experiments incorporating normalized latitude and longitude fields as inputs did not yield significant improvements in model performance. There was also a distinct correlation between the complexity of the CNN architecture and its performance, with deeper networks achieving superior outcomes and the U-net-based architectures outperformed other models. Our initial comparison between Simple CNNs and the U-Net architecture provided valuable insights into the importance of network complexity and the use of skip connections in capturing the spatial features necessary for accurate data assimilation emulation. Although Simple CNNs are less commonly used in current geoscience applications, this analysis was important in demonstrating the necessity of employing advanced architectures like U-Net for such complex tasks.

Our findings also indicated that even a shallow CNN with a simple architecture could assimilate SSH observations with an error margin of only 8 mm, compared to the T-SIS assimilation package. Additionally, experiments assessing the impact

of land on ocean models revealed that CNNs remained robust against land by simply zeroing out these regions, not affecting the models' performance based on the percentage of ocean used in the training data. Moreover, the experiments showcased the CNNs' ability to efficiently handle additional inputs without performance degradation and to assimilate multiple fields simultaneously. Another important aspect highlighted through our study is the importance of selecting appropriate test sets to evaluate the generalization capabilities of deep learning models, particularly when dealing with realistic ocean models over shorter time scales such as weeks or months. Using a random selection of training data as a test set could lead to misleadingly favorable results if the oceanographic conditions do not change significantly.

Data leakage is a critical issue in machine learning applications within Earth sciences due to the temporal and spatial dependencies in the data. Our approach of chronological data splitting and testing on entirely separate years aims to mitigate this concern. To test the generalization of our proposed model, we utilized data from two different years that presented varied dynamical states of the GoM. Although errors were slightly higher than with the initial test data (maybe due to the proximity of the training and test sets), an error of 4 mm was observed as a typical value when applying our CNN data assimilation (DA) method, and this is the expected error of our model in operational systems.

Furthermore, we compared the time performance of a traditional DA method (optimal interpolation), implemented in FORTRAN and executed on High-Performance Computing clusters, with our proposed CNN method running on a single GPU. These comparisons, while challenging, provided insights into potential time and cost savings achievable with new technologies. In our tests, the CNN model approximated the DA optimal interpolation method with less than a 4 mm error for SSH and achieved potential speedups of up to 58 times compared to systems running on a 32-processor cluster.

Despite ongoing research into explainable AI, which aims to better understand decisions made by deep learning models, these techniques typically do not analyze specific performance comparisons related to model design decisions. This work offers insights into the expected behavior of CNNs when applied to the specific problem of data assimilation in ocean models. Most findings from the proposed experiments should also be applicable in other scenarios where CNN models are used for 'image-to-image' modeling in oceanic and atmospheric predictions involving geographic coordinates and diverse fields.

While our study focused on traditional CNN architectures, specifically U-Nets, we acknowledge that newer deep learning techniques such as attention-based models (e.g., CBAM in SmaAt U-Nets), Vision Transformers (ViT), and Denoising Diffusion Models have demonstrated superior performance in various image processing tasks by preserving fine details and reducing smoothing effects. Incorporating these advanced architectures into ocean data assimilation represents a promising direction for future research. However, due to computational constraints and the scope of this study, we did not explore these models. Future work will aim to investigate these techniques, leveraging their strengths to further enhance the accuracy and efficiency of data assimilation in ocean models.

*Author contributions.* **Olmo Zavala-Romero**: Conceived and designed the experiments, ran all the machine learning trainings, analyzed the data, and wrote the majority of the paper. **Alexandra Bozec**: Generated the data for training the models; configured the ocean model, the assimilation package, and ran the assimilated HYCOM for all the years. Reviewed drafts of the paper. **Eric P. Chassignet**: Conceived the

general design of the research, analyzed the data, and reviewed drafts of the paper. **Jose R. Miranda**: Analyzed the data, wrote sections of the paper, and reviewed drafts of the paper.

*Competing interests.* The authors declare that they have no competing interests.

*Acknowledgements.* We would like to express our gratitude to the Office of Naval Research for their partial support under grant N00014-20-1-2023 (MURI ML-SCOPE). We acknowledge the use of artificial intelligence tools which assisted in the editing of this manuscript.

*Code availability.* The source code supporting the findings of this study is openly available in the "da_hycom" repository on GitHub, hosted at https://github.com/olmozavala/da_hycom. The repository contains all necessary scripts required for implementing the models and algorithms discussed in this paper. Users can download, fork, or contribute to the project under the terms of the license specified within the repository.

The data used for training the models are available from the HYCOM (Hybrid Coordinate Ocean Model) website. Interested readers can access and download the training data by visiting the HYCOM project page at https://www.hycom.org. For any issues or further inquiries related to the code, please open an issue directly on the GitHub repository page.

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
