# Peer review of "Convolutional Neural Networks for Sea Surface Data Assimilation in Operational Ocean Models: Test Case in the Gulf of Mexico"

_EGUsphere, 2024_

## Author Comment (AC1)

We would like to express our sincere gratitude for the time and effort you have dedicated to reviewing our manuscript. Your insightful comments and constructive feedback have been invaluable in enhancing the clarity, rigor, and overall quality of our work. Each of your suggestions has provided us with the opportunity to refine our methodologies, clarify our explanations, and ensure that our findings are presented in the most comprehensive and accessible manner possible.

We deeply appreciate your attention to detail and your commitment to advancing research in this field. Your expertise has significantly contributed to the improvement of our study, and we are confident that the revisions made in response to your feedback have strengthened our manuscript. Thank you once again for your thoughtful and thorough reviews.

Best regards,

Olmo Zavala-Romero
* * *
**In their work, the Authors address one very important initial step for making learned GCMs operational, as well as a way to accelerate a costly process in operational geosciences with traditional GCMs, namely the assimilation of observations into the operational framework. Given the chaotic nature of the system, any numerical model, learnt or not, requires updating with real observations, but exactly matching sparse and noisy real observations is a nonoptimal solution to the operational process.**

**My issues with the paper are both structural and scientific.**

**The authors remain unclear in the abstract as well as prior to the experiment phase as to the input and outputs of the deep learning models used. Namely, they should clarify early on that they train a Deep Learning Architecture to reproduce the outputs of the T-SIS Assimilation, given the model forecast of SST and SSH and the simulated (? that has stayed unclear) satellite observations of SST and SSH. I sincerely hope that this is what is happening, since it remains unclear to me. I searched through the manuscript for an explanation of the observations, and it is nowhere to be found.** *Thank you for your comments and for highlighting the lack of clarity regarding the inputs and outputs of our deep learning models. You are correct in your understanding. Our CNN models are trained to replicate the increments generated by the T-SIS data assimilation package, given:*

*Inputs: Real satellite observations of SST from GHRSST, along-track altimeter SSH observations (ADT), the model background state (previous forecast), and the innovations (differences between observations and background).*

*Outputs: The increments (corrections) that should be applied to the model forecast to assimilate the observations.*

*We have revised the abstract to clearly state this and added a detailed explanation in the "Data" subsection in the manuscript.*

**It is implied this is a twin experiment, but it really warrants clarification.** *Thank you for your comment and for pointing out the need to clarify the nature of our experiments. We apologize for any confusion caused by the implication that our study is a twin experiment.*

*Our study is not a twin experiment. In a traditional twin experiment, synthetic observations are generated from a model run (often referred to as the "truth"), and these observations are then assimilated back into the model to evaluate the data assimilation system's ability to recover the known truth under controlled conditions.*

*In contrast, our experiments use real observational data:*

*- Real satellite observations of sea surface temperature (SST) from the Group for High Resolution Sea Surface Temperature (GHRSST). - Real along-track altimeter observations of sea surface height (SSH) from satellite missions. - The model background state (previous forecast) from HYCOM. - Innovations, calculated as the differences between the observations and the model background state.*

*Outputs:*

*- The increments generated by the T-SIS data assimilation system, representing the corrections applied to the model forecast to assimilate the real observations.*

*Our Convolutional Neural Network (CNN) models are trained to replicate the increments produced by T-SIS using these real observations and model states. The goal is to assess whether CNNs can emulate the data assimilation step performed by T-SIS when provided with actual observational data, rather than synthetic data derived from the model itself.*

*Moreover, our CNN models are not yet implemented as a full replacement for the data assimilation process in HYCOM. Instead, they serve as a proof-of-concept to demonstrate the potential of machine learning to approximate the corrections made by traditional data assimilation methods using real-world data.*

*We have added the following paragraph to explicitly state the nature of our experiments:*

*"It is important to note that our experiments are not twin experiments. In twin experiments, synthetic observations are generated from a model run (considered the "truth") and are then assimilated back into the model to assess the data assimilation system under controlled conditions. In our study, we utilize real observational data for both training and testing our CNN models. The T-SIS data assimilation system generates increments based on these real observations, and our CNN models are trained to replicate these increments. By using actual observations from GHRSST for SST and along-track altimeter data for SSH, our experiments reflect a realistic scenario where the CNN models learn from real-world data, capturing the complexities and uncertainties inherent in operational ocean modeling."*

**Similarly the assimilation step seems to be daily, but the assimilated field is complete, which is wildly unrealistic given the fields used as inputs.** *Thank you for your comment regarding the assimilation step and the completeness of the assimilated fields in our study.*

*You are correct that our assimilation step is performed daily. Regarding the nature of the assimilated fields and the inputs used:*

*Sea Surface Temperature (SST): The SST observations we assimilate come from the Group for High Resolution Sea Surface Temperature (GHRSST) dataset. This dataset provides interpolated SST fields with complete spatial coverage.*

*Sea Surface Height (SSH): The SSH observations are derived from along-track altimeter data (ADT tracks), which are indeed sparse and discontinuous in both space and time. While we resample these observations onto the model grid using nearest-neighbor interpolation, they remain sparse and do not form a complete field on their own.*

*A key aspect of the data assimilation (DA) approach—both in the traditional T-SIS method and our CNN models—is the ability to propagate the information from sparse observations throughout the model domain to produce a complete assimilated field.*

*To address your concern and enhance the clarity of our methodology, we have added an additional figure (now Figure 2 in the revised manuscript) that illustrates the inputs and outputs of our model during the assimilation process. This figure displays:*

*Possible inputs:*

*- The sparse SSH observations (ADT tracks) overlaid on the model grid. - The complete SST observational field from GHRSST. - The model background state of SSH and SST (prior forecast). - The model innovations (difference between the observations and the model) for SST and SSH.*

*Additional inputs to improve the prediction:*

*- 200m mask - Normalized latitude and longitude*

*Outputs: The desired increments for the SSH and SST fields.*

*We have also expanded the explanation in the "Data" section to clarify this point:*

*"It's important to note that while the SST observations from GHRSST provide near-complete spatial coverage, the SSH observations from along-track altimeter data are sparse and irregularly distributed. The DA schemes are able to handle such sparse datasets and propagate the observational information across the model domain. This is achieved through statistical interpolation and the physical dynamics represented in the model by the T-SIS system, which together allow us to estimate the ocean state in unobserved areas based on the available observations."*

**As to the rest of the article, the techniques used are now ~10 year old approaches, lacking a lot of significant improvements from the architectural side, but more importantly, on the implementation side, there are lots of elements missing:**

**The pre-processing is not detailed. Given the different value ranges of the input variables, it is crucial to perform a normalization beforehand.** *Thank you for highlighting the importance of detailing the preprocessing steps in our manuscript. We agree that normalization is a crucial aspect when training machine learning models, especially with input variables that have different value ranges.*

*We have added a dedicated Preprocessing section to the manuscript to thoroughly describe the steps we have taken to prepare the data for training our CNN models.*

*Added Preprocessing section:*

*Prior to training the Convolutional Neural Network (CNN) models, we performed several preprocessing steps to ensure that the input data was appropriately scaled and formatted. First, to address the issue of differing value ranges among the input variables, we normalized each field individually. This normalization involved adjusting each input field—such as sea surface temperature (SST) observations and sea surface height (SSH) observations—to have a mean of zero and a standard deviation of one. Normalization is an important step in machine learning, as it ensures that all input features contribute equally during training, preventing variables with larger magnitudes from disproportionately influencing the model's learning process. By standardizing the inputs, we facilitated a more stable and efficient optimization during model training.*

*The parameters used for normalization, specifically the mean and standard deviation for each input field, were calculated using the data from the full training period, encompassing the years 2009 and 2010. These calculated parameters were then applied to the validation and test datasets, as well as to the additionally tested years 2002 and 2006.*

*After the CNN models generated the predicted increments, we applied an inverse transformation using the previously calculated mean and standard deviation to denormalize the outputs. This denormalization step converted the increments back to their original units—such as degrees Celsius for SST or meters for SSH—making them compatible with the model forecast corrections. By restoring the original scale of the data, we ensured that the increments could be directly applied to the HYCOM model outputs.*

*We addressed the irregular distribution and missing values of the along-track altimeter SSH data by mapping these observations onto the model grid and filling the missing data points with zeros. Representing the absence of observations with zeros allowed the CNN models to process the SSH data as continuous fields, where zeros explicitly indicated locations without observational data. The response of the CNN to missing values represented as zeros is of interest to us and was part of the experiments.*

**Missing values: how are the missing values handled? The lack of mention of missing values seems to indicate a twin experiment, and even then it is unrealistic given the nature of the data.** *Thank you for bringing up this important point regarding the handling of missing values in our study.*

*In our experiments, we work with real observational data, particularly the sea surface height (SSH) observations from along-track altimeter measurements, which are inherently sparse and contain missing values due to the satellites' orbit paths and revisit times. These missing values are a realistic characteristic of satellite observations and present a common challenge in operational data assimilation.*

*To incorporate these sparse observations into our Convolutional Neural Network (CNN) models, we mapped the SSH observations onto the model grid and represented missing values by filling them with zeros. This approach allowed the CNN models to process the SSH data as continuous fields, where zeros explicitly indicated locations without observational data.*

*We have added a detailed explanation of how missing values are handled in the newly included Preprocessing section of the manuscript.*

**There is no care taken for data leakage. There are no dates dropped between train validation and test.** *Thank you for bringing up this critical point regarding data leakage. We agree that data leakage is a significant concern in machine learning applications, particularly in Earth sciences and even more in the ocean, where temporal autocorrelation is prevalent, and the state of the ocean does not vary significantly on a daily basis.*

*To enhance the clarity regarding our handling of data leakage, we have made the following modifications:*

*Inside Data Splitting and Methodology: In Earth sciences, particularly in ocean modeling, data leakage is a significant concern due to the strong temporal autocorrelation in the data. The state of the ocean does not change dramatically over short periods, which means that random splitting of data can lead to leakage where the model learns from future information. To mitigate this, we employed a chronological data splitting strategy. From the 730 daily examples the first 80% is used for training, 10% for validation, and the last 10% is used for testing, ranging from October $19^{\text{th}}$ to December $31^{\text{st}}$ of 2010.*

*This method ensures that the model is trained on past data and evaluated on future data, reducing the risk of information from the test set influencing the training process. However, we recognize that the proximity of the*

*training and test sets may still allow for some data leakage due to the ocean's slow-changing nature.*

*To further assess the model's ability to generalize and to address potential data leakage, we tested the model on datasets from the years 2002 and 2006. These years were selected because they exhibit different dynamical states of the GoM, with the Loop Current mostly in retracted and extended phases, respectively. By evaluating the model on data that is entirely separate from the training and validation sets and representing different oceanographic conditions, we reduce the likelihood that the model's performance is artificially inflated due to data leakage.*

*The model maintained strong performance on these additional datasets, with RMSE values comparable to those on the original test set as described in the Generalization Tests section.*

*Modified Conclusions:*

*Data leakage is a critical issue in machine learning applications within Earth sciences due to the temporal and spatial dependencies in the data. Our approach of chronological data splitting and testing on entirely separate years aims to mitigate this concern. To test the generalization of our proposed model, we utilized data from two different years that presented varied dynamical states of the GoM. Although errors were slightly higher than with the initial test data (maybe due to the proximity of the training and test sets), an error of 4 mm was observed as a typical value when applying our CNN data assimilation (DA) method, and this is the expected error of our model in operational systems.*

**The layers do not include any form of regularisation: Dropout, AdamW with a heavier weight loss penalization of (my preference) Batch Normalization** *Thank you for your comment regarding the use of regularization in our neural network models. We appreciate the opportunity to clarify our approach and address your concerns.*

*Clarification on Regularization Techniques Used:*

*Batch Normalization:*

*We did incorporate Batch Normalization layers after each convolutional layer in our U-Net architecture, except for the final output layer. Batch Normalization is known to stabilize and accelerate the training process by normalizing the inputs to each layer, reducing internal covariate shift. It also provides a form of regularization by mitigating the risk of overfitting.*

*We acknowledge that this detail was not explicitly stated in the original manuscript. To rectify this oversight, we have updated the corresponding figure (now Figure 4) to illustrate the inclusion of Batch Normalization layers. Additionally, we have added a description in the text to highlight their use in our U-Net models.*

*Dropout:*

*We conducted initial experiments incorporating a Dropout rate of 20% after the convolutional layers. However, we observed that the performance of the networks with Dropout was lower than that of networks without Dropout. We hypothesize that this decrease in performance may be due to our training dataset not being large enough for Dropout to be effective. In smaller datasets, Dropout can sometimes hinder learning by excessively reducing the network's capacity during training. Therefore, we decided not to include Dropout in our final models to maintain optimal performance.*

*Optimizer Choice (AdamW):*

*We appreciate your suggestion regarding the use of the AdamW optimizer with increased weight decay as a form of regularization through L2 weight penalization. In our experiments, we utilized the standard Adam optimizer with default parameters. We agree that employing AdamW could potentially improve generalization by providing stronger regularization. We plan to explore the use of AdamW with a higher weight decay coefficient in future work to assess its impact on model performance.*

**Newer techniques of deep learning are not addressed: CBAM layers in Sma-at U-Nets, or Masked Autoencoder ViT, or Denoising Diffusion Inpainting which are known to outperform U-Nets which tend to smooth out the output field** *Thank you for your comment and for highlighting newer deep learning techniques such as the Convolutional Block Attention Module (CBAM) in SmaAt U-Nets, Masked Autoencoder Vision Transformers (ViT), and Denoising Diffusion Inpainting models. We appreciate your suggestion to consider these advanced architectures, which have demonstrated superior performance in preserving fine details and reducing the smoothing effect often associated with traditional U-Nets.*

*The primary objective of our study was to explore the feasibility of using Convolutional Neural Networks (CNNs), specifically U-Net architectures, to emulate the data assimilation process performed by the T-SIS optimal interpolation method in an operational ocean model setting. We aimed to establish a foundational understanding of how deep learning models can be applied to this domain before delving into more complex architectures.*

*While we recognize the potential benefits of incorporating these newer techniques, integrating such advanced models into our current experimental pipeline presents significant computational challenges. We have trained a total of 75 CNN models in our experiments, which is computationally intensive. Incorporating additional models would require a substantial amount of computational resources that are currently beyond our capacity.*

*Additionally, our study serves as a proof-of-concept to demonstrate that CNN-based models can approximate the data assimilation increments generated by T-SIS. By focusing on U-Net architectures, we aimed to build a solid baseline and understand the fundamental capabilities and limitations of CNNs in this context before exploring more sophisticated models.*

*We agree that incorporating attention mechanisms like CBAM, exploring Vision Transformers, and utilizing Denoising Diffusion Models could potentially enhance the performance of data assimilation models by preserving finer details and reducing smoothing effects. These techniques are indeed promising and relevant to our research area.*

*To acknowledge the importance of these newer techniques and outline our plans for future research, we have added the following paragraph to the Conclusions section:*

*"While our study focused on traditional CNN architectures, specifically U-Nets, we acknowledge that newer deep learning techniques such as attention-based models (e.g., CBAM in SmaAt U-Nets), Vision Transformers (ViT), and Denoising Diffusion Models have demonstrated superior performance in various image processing tasks by preserving fine details and reducing smoothing effects. Incorporating these advanced architectures into ocean data assimilation represents a promising direction for future research. However, due to computational constraints and the scope of this study, we did not explore these models. Future work will aim to investigate these techniques, leveraging their strengths to further enhance the accuracy and efficiency of data assimilation in ocean models."*

**The network in all its configurations is not provided with any information on latitude or longitude, therefore preventing it from knowing contextually the Coriolis force** *Thank you for your comment*

*regarding the inclusion of latitude and longitude information in our neural network models. We agree that providing the network with spatial coordinates could enhance its ability to capture geophysical processes influenced by location, such as the Coriolis force, which varies with latitude.*

*Recognizing the importance of spatial context, we conducted additional experiments where we included normalized latitude and longitude fields as input channels to the network. The latitude and longitude values were normalized to a range between 0 and 1 to maintain consistency with the scaling of other input features. These experiments were performed specifically on the set that varied the window size (box size), resulting in an additional 20 models trained and evaluated.*

*Results and Observations:*

*Interestingly, the inclusion of latitude and longitude did not significantly change the trends observed in our original experiments. The models' performance remained consistent with the previous results, indicating that the spatial coordinates did not provide substantial additional information for the network to improve its predictions in this context. This suggests that the network may already be capturing location-dependent information implicitly through the input data provided, such as sea surface temperature (SST) and sea surface height (SSH) fields.*

*We have included a description of the additional experiments where normalized latitude and longitude fields were incorporated into the models. The text now reads:*

*"To assess the impact of spatial coordinates on model performance, we conducted additional experiments by including normalized latitude and longitude fields as input channels to the network. The latitude and longitude values were scaled between 0 and 1 to align with the normalization of other input features."*

*We have updated the results to include findings from the models that incorporated latitude and longitude. The revised text states:*

*"The inclusion of latitude and longitude as additional inputs did not significantly alter the performance of the models across different window sizes as shown in Figure 6. The root mean square error (RMSE) and other performance metrics remained comparable to those of the models without spatial coordinates, indicating that the network did not benefit from the explicit addition of latitude and longitude in this context."*

*In the conclusions, we have discussed the implications of these findings and potential future research directions:*

*"Our experiments incorporating normalized latitude and longitude fields as inputs did not yield significant improvements in model performance."*

**As far as the experiments are concerned, the presentation and analysis of the multiple base CNNs which are not really in use nowadays for these types of problems do not seem useful. Running this experiment with U-Nets that learn over different patches could be interesting, potentially.** *Thank you for your insightful feedback on our manuscript. We apologize for any confusion regarding the use of the 'Simple CNN' architectures in our experiments.*

*The 'Simple CNN' architectures were utilized only in one of the experiments, which aimed to compare different network complexities and assess the impact of model depth and capacity on performance. Our intention was to establish a baseline and understand how simpler architectures perform relative to more advanced models.*

All subsequent experiments, including those involving different patch sizes (window sizes), input configurations, ocean coverage percentages, and generalization tests across different years, were conducted using the U-Net architecture exclusively.

By evaluating models of varying complexity, we aimed to establish a performance baseline. This helps in understanding whether the added complexity of U-Nets is justified by a significant improvement in performance. Comparing Simple CNNs with the U-Net allowed us to investigate how network depth and the presence of skip connections affect the model's ability to replicate the data assimilation increments. The results from this comparison highlight the advantages of using more sophisticated architectures like U-Net for complex geoscientific tasks, thereby justifying their use in subsequent experiments.

To enhance clarity and avoid any misunderstanding, we have made the following revisions:

We have clarified the scope of the Simple CNNs:

"In the experiment analyzing network complexity, we evaluated different network complexities by comparing Simple CNN architectures with varying depths (2, 4, 8, and 16 layers) to the U-Net architecture. This experiment aims to assess the impact of network depth on model performance. All subsequent experiments utilize the U-Net architecture exclusively to explore the effects of window size, input configurations, ocean percentage, etc."

We have included a paragraph explaining the inclusion of Simple CNNs:

"Our initial comparison between Simple CNNs and the U-Net architecture provided valuable insights into the importance of network complexity and the use of skip connections in capturing the spatial features necessary for accurate data assimilation emulation. Although Simple CNNs are less commonly used in current geoscience applications, this analysis was crucial in demonstrating the necessity of employing advanced architectures like U-Net for such complex tasks."

**The results however are encouraging and should the Authors significantly expand and clarify their paper, I would consider it a worthy contribution to the field.** *Thank you for your encouraging remarks regarding our results. We appreciate your recognition of the potential contribution our work can make to the field. In response to your valuable feedback, we have significantly expanded and clarified our paper to address the points you raised.*

Specifically, we have:

- *Enhanced Clarity: Clarified our experimental design, methodology, and the handling of real observational data.*
- *Detailed Preprocessing Steps: Provided thorough explanations of data normalization and the handling of missing values.*
- *Addressed Data Leakage: Explained our data splitting strategy and conducted additional tests to assess model generalization.*
- *Expanded Literature Review: Included references to relevant U-Net applications in geoscience over the past decade.*
- *Improved Presentation: Updated figures and tables for better clarity and understanding.*
- *Considered Advanced Techniques: Discussed newer deep learning methods and outlined plans for future research.*

We appreciate your guidance, which has helped us improve the manuscript. We hope the revised paper meets your expectations and contributes meaningfully to the field.

// Minor comments:

**Section 2.2 would benefit from a quick bibliographical referencing of some of the many U-Nets and Sma-at U-Nets applied in a multitude of geoscience problems over the last 10 years.** *Thank you for your insightful suggestion. We agree that including a brief overview of U-Net architectures and their applications in geoscience over the past decade would enhance Section 2.2 and provide valuable context for our work. U-Nets and their variants have indeed been widely applied in various geoscientific fields, demonstrating their effectiveness in handling spatial data and complex patterns similar to those in our study.*

*We have revised Section 2.2 to include a discussion of notable applications of U-Nets in geoscience, along with appropriate references to key studies. This addition highlights the relevance of U-Net architectures in addressing problems related to Earth observation, remote sensing, climate modeling, oceanography, and other areas within geoscience.*

*Revised Section 2.2:*

*Over the past decade, U-Net architectures have been extensively applied to a variety of geoscience problems due to their capability to learn hierarchical features and capture both local and global contexts. Variants of U-Net, such as the attention U-Net (ref) and the Small Attention-UNet (SmaAt-UNet) (ref), have been developed to enhance feature extraction and improve performance in complex geoscientific tasks. These variants introduce mechanisms like attention gates and efficient channel interdependencies, allowing models to focus on relevant features while reducing computational requirements. Some notable applications of U-Nets in geoscience include:

- **Remote Sensing and Earth Observation:** U-Nets have been extensively used for semantic segmentation and classification of satellite imagery, including land cover mapping (ref), building and road extraction (ref), and change detection (ref).
- **Meteorology and Climate Science:** U-Net architectures have been employed for precipitation nowcasting using radar data (ref).
- **Hydrology and Flood Mapping:** U-Nets have been applied to flood detection and mapping from satellite images (ref), and mountain ice segmentation (ref).
- **Oceanography:** U-Net architectures have been utilized in oceanography for bathymetry estimation from optical imagery (ref), and ocean eddy detection and classification (ref).
- **Data Assimilation and Ocean Modeling:** (ref) introduced Multimodal 4DVarNets, where U-Net-based architectures obtain similar results to 4DVarNets for the reconstruction of sea surface dynamics by leveraging synergies between sea surface temperature (SST) and sea surface height (SSH) observations. Their work demonstrates the capability of deep learning models to assimilate multiple data modalities and reconstruct ocean surface variables with high accuracy.

*The versatility of U-Net architectures in geoscientific applications makes them a suitable choice for data assimilation in ocean modeling, given their ability to capture spatial dependencies and manage multiscale features. This capability aligns well with the demands of integrating observational data into ocean models, motivating our choice to adopt this architecture.*

**Consider flipping table to horizontally into a two-column paradigm so as to not imply linewise combinations of parameters.** *Thank you for your insightful suggestion regarding the presentation of Table 2. We understand that the current format of the table, which lists parameters in columns and combinations in rows, might inadvertently imply that only the linewise combinations of parameters were tested in our experiments.*

*We have updated Table 2 in the manuscript to reflect this new format. Additionally, we have revised the accompanying text to ensure consistency and to explain that the experiments involved testing various combinations of these parameters independently.*

*New text: Table (ref) summarizes the parameters and their respective values tested in our experiments. Each parameter was varied independently to assess its impact on the performance of the CNN models in assimilating oceanographic data. By exploring different combinations of window sizes, CNN complexities, ocean percentages, inputs, and outputs, we aimed to gain comprehensive insights into the behavior and capabilities of the models under various conditions. Each tested model is trained five times to gather statistics on the training's consistency and allow a more accurate comparison between the models' performances. A total of 75 CNN models are evaluated in these experiments.*

---

## Author Comment (AC2)

**Response to Reviewer 2**

We would like to express our sincere gratitude for the time and effort you have dedicated to reviewing our manuscript. Your insightful comments and constructive feedback have been invaluable in enhancing the clarity, rigor, and overall quality of our work. Each of your suggestions has provided us with the opportunity to refine our methodologies, clarify our explanations, and ensure that our findings are presented in the most comprehensive and accessible manner possible.

We deeply appreciate your attention to detail and your commitment to advancing research in this field. Your expertise has significantly contributed to the improvement of our study, and we are confident that the revisions made in response to your feedback have strengthened our manuscript. Thank you once again for your thoughtful and thorough reviews.

Best regards,

Olmo Zavala-Romero
* * *
**This work explores the ability of Convolutional Neural Networks (CNNs) to serve as a surrogate to the Tendral Statistical Interpolation System (TSIS) method of data assimilation between the Hybrid Coordinate Ocean Model (HYCOM) and observations. In addition to evaluating the performance of their models in the Gulf of Mexico, the authors also quantify the difference in skill of various hyperparameters and model architectures in this highly dynamic region. The results of this paper show the technique is sound and has potential to be used operationally.**

**While the study is well conducted and the results are relevant to the community, the report will benefit from more specificity in the model architectures and data preprocessing. All such information is discernible in the code repository provided but should be expressed in the report. This includes:**

- Preprocessing
- The handling of land points (e.g., masked as 0)
- Structure of tensors used as input
- Use of batch normalization
- Point where normalization/denormalization occurs
- Structure of output tensors
- Which parameters were tuned via hyperparameter optimization

*Thank you for your thoughtful comments and for recognizing the relevance of our study. We appreciate your suggestion to provide more detailed information on the model architectures and data preprocessing in the manuscript. We agree that including these specifics will enhance the clarity and reproducibility of our work.*

*In response to your feedback, we have thoroughly addressed each of your points below and have made corresponding revisions to the manuscript.*

*We have added a new subsection titled "Data Preprocessing" in Section 3.2 of the manuscript, where we detail the entire preprocessing steps applied to our data before training the CNN models.*

*Summary of the preprocessing steps:*

- *We use daily outputs from the assimilative HYCOM model for the years 2009 and 2010. The data includes background state variables, observation fields, and increment fields (the corrections applied by T-SIS). Each input variable is normalized to have zero mean and unit standard deviation (mean 0, standard deviation 1). The mean and standard deviation are computed from the training set (2009 and 2010). This normalization is applied consistently to the test sets (last 10% of 2010 and the years 2002 and 2006).*

- *Missing values in the input data (e.g., due to land areas or lack of observations) are filled with zeros after normalization. The loss function is computed only over ocean grid points. By masking out land points during loss computation, the network focuses on learning the ocean dynamics and does not penalize predictions over land.*

- *We include an input layer mask indicating areas with depths greater than 200 meters. Since T-SIS does not provide SSH increments for shallow areas (<200m), we provide this mask as an input to the CNN to help it learn this restriction. After the CNN makes predictions, we apply the land mask to set the values at land grid points to NaN.*

- *The input tensors to the CNN models are four-dimensional arrays with the following dimensions: [Batch Size, Height, Width, Channels]. The numbers for Height, Width, and Channels vary depending on the experiment. All input tensors are of type float32.*

*We have added this description of the input tensor structures in the "Data Preprocessing" section.*

*Batch Normalization is used in our U-Net architecture to improve training stability and performance. Batch Normalization layers are applied after each convolutional layer and ReLU activation in the U-Net architecture, except for the final output layer. This is reflected in the updated Figure 2, which now includes Batch Normalization layers. We have updated the description of the U-Net architecture in Section 3.3 to include this information.*

*We did not perform extensive hyperparameter optimization. The hyperparameters used were chosen based on standard practices and preliminary testing: We used the Adam optimizer with a learning rate of 0.001 (1e-3). This learning rate provided stable convergence across our experiments. A batch size of 32 was used for all models. Training was monitored using validation loss. We implemented early stopping if the validation loss did not improve for 20 consecutive epochs. The model with the lowest validation loss was selected for evaluation.*
* * *
**Finally, there should be some comparison to other models/techniques used for similar purposes in geosciences. Much has been written on this topic in the atmospheric sciences. How could this study be extended to newer, more sophisticated model architectures?**

*Thank you for your comments and for highlighting the importance of situating our study within the broader context of geoscientific modeling techniques. We agree that comparing our approach with other models and techniques, particularly those prevalent in ocean sciences, would enhance the comprehensiveness and relevance of our work.*

*We have included a discussion on the potential integration of advanced architectures such as Convolutional Block Attention Modules (CBAM) in SmaAt U-Nets, Masked Autoencoder Vision Transformers (ViT), and Denoising Diffusion Inpainting models. Given that CNNs remain the backbone for some of these more advanced models, our findings regarding CNN performance and data assimilation are expected to extend to these.*

*Specifically, the principles of spatial feature extraction, suggesting that our insights on window size and input configurations would still be applicable.*

*We have also included in the conclusions the scope of our work and potential future research directions with newer techniques.*
* * *
**Line 80: The "Markov process" mentioned is not described or referenced. Presumably, modelers and computational scientists will be familiar with the meaning, but it wouldn't hurt to briefly describe this meaning.**

*Thank you for your comment. We agree that providing a brief description of the Markov process will enhance the clarity of our manuscript. We have revised the relevant section to include a concise explanation and an appropriate reference.*

*Revised Text:*

*"In this approach, it is assumed that the model forecast follows a Markov process, which means that the future state of the system depends only on its current state and not on any previous states (Davis, 2013). Observations can improve the estimate of the model state in a least squares sense, taking into account the modeled and observed error covariances as follows."*
* * *
**Line 98: Similar to above; Gaussian Markov Random Field not explained or referenced.**

*Thank you for your valuable feedback. We agree that providing a brief explanation of the Gaussian Markov Random Field (GMRF) will enhance the clarity and comprehensiveness of our manuscript. We have revised the relevant section to include a concise description of GMRF and an appropriate reference.*

*Revised Text:*

*"The information matrix is modeled using a Gaussian Markov Random Field (GMRF), which is a probabilistic model consisting of a set of random variables having a multivariate Gaussian distribution, with the Markov property that each variable is conditionally independent of all others given its immediate neighbors (Rue and Held, 2005)."*
* * *
**Sect. 2.2: If CNNs are going to be explained in this detail, it would help to show a figure differentiating traditional CNNs from UNets/encoder-decoder networks since the difference is hard to visualize. UNets are referenced, CNNs are not. There should be consistency in the degree of explanations in this section.**

*Thank you for your feedback regarding Section 2.2 of our manuscript. We agree that providing a visual comparison between traditional Convolutional Neural Networks (CNNs) and U-Net/encoder-decoder architectures would enhance the reader's understanding of the structural differences and functional advantages of these models.*

*To address your suggestion, we have incorporated the classical LeCun et al. (1998) reference for traditional CNNs to acknowledge their foundational role in deep learning. Additionally, we have included a new figure that clearly differentiates traditional CNN architectures from U-Net/encoder-decoder networks, highlighting key components such as the encoder, decoder, and skip connections inherent to U-Nets.*

*Furthermore, we have expanded the description of U-Net architectures to ensure consistency in the level of detail provided for both CNNs and U-Nets, thereby offering a more balanced and comprehensive explanation of each model type.*
* * *
**Line 156: "hindcast" is used to describe the training data (I assume). There is no mention of performing the TSIS technique with awareness of time, so can this be called a hindcast?**

*Thank you for your observation regarding the use of the term "hindcast" in our manuscript. You are correct that "hindcast" may not be the most appropriate term in this context. While we are indeed predicting past states, the TSIS technique employed does not utilize observations beyond the current time step, which is part of hindcasting.*

*To address this, we have revised the manuscript to replace "hindcast" with "analysis."*
* * *
**Line 150: I am not familiar with the use of the word "innovations" here and in the following line. Could you rephrase?**

*Thank you for your feedback. We understand that the term "innovations" may not be universally familiar outside the data assimilation community. To enhance clarity, we have revised the manuscript to define "innovations" explicitly.*
* * *
**Line 291: Why this day specifically? Whether it was randomly selected or chosen based on best/worst performance should be noted.**

*Thank you for pointing this out. The day was selected randomly to represent a typical example of our model's performance. We will update the manuscript to clarify that this day was chosen at random to showcase the model behavior.*

*Revised Text:*

*"This day was randomly selected to provide a representative example of the model's performance in a typical scenario."*
* * *
**Line 347: Units should be placed on "58" and "22". It is unclear that these are factors until reading the table.**

*Thank you for your feedback. The speedup factors "58" and "22" are unitless ratios representing the performance improvements of our CNN model compared to the T-SIS method. To enhance clarity, we have added a descriptive explanation in the manuscript. This clarification ensures that the context and meaning of the speedup factors are immediately apparent.*

*Added Text:*

*"The speedup or speedup factor is a unitless measure defined as the ratio of the time taken by the traditional T-SIS method to the time taken by our proposed CNN model for the same assimilation task. For example, a speed-up factor of 58 indicates that the CNN model performs the assimilation 58 times faster than the T-SIS method running on a 32-processor cluster, while a speed-up factor of 22 signifies a 22-fold increase in performance compared to the 96-processor T-SIS configuration."*

**Technical Comments**

**Throughout the paper, citations are not delineated from the clause that precedes them (i.e., separated with a comma or wrapped in parentheses). Some examples: Line 48-49, Line 117-118, Line 126, Line 144, Line 153, Line 219.**

*Thank you for your attention to the citation formatting in our manuscript. We understand the importance of clear and consistent citation practices for enhancing the readability and professionalism of our work. In response to your comment, we have thoroughly reviewed the manuscript and revised all instances where citations were not properly delineated from the preceding clauses. Specifically, we have ensured that all citations are appropriately wrapped in parentheses and, where necessary, separated by commas to maintain clarity.*

**Line 114: "coming" → "coming"**

*We reviewed this comment, but could not find the error in the document. If there is a specific instance that still needs correction, we would appreciate any further clarification.*

**Line 125: "U-net" should be plural.**

*Thank you for this suggestion. We have revised the text to use the plural form "U-Nets" where appropriate to maintain consistency.*

**Line 156: "test" → "testing"**

*Thank you for pointing this out. We have corrected the text to use "testing" instead of "test" to maintain proper terminology.*

**Line 197: "1[h]"?**

*Thank you for bringing this to our attention. We have fixed this issue by providing the correct time unit. The text has been updated for clarity.*

We appreciate your thorough review and constructive feedback, which have helped us improve the quality and clarity of our manuscript. We hope the revised version addresses all of your concerns and contributes meaningfully to the field.